# FaithShield: Defending Vision–Language Models Against Explanation Manipulation via X-Shift Attacks

## Abstract

Vision–Language Models (VLMs) such as Contrastive Language–Image Pretraining (CLIP) have achieved remarkable success in aligning images and text, yet their explanations remain highly vulnerable to adversarial manipulation. Recent findings show that imperceptible perturbations can preserve model predictions while redirecting heatmaps toward irrelevant regions, undermining the faithfulness of the explanation. We introduce the X-Shift attack, a novel adversarial strategy that drives patch-level embeddings toward the target text embedding, thereby shifting explanation maps without altering output predictions. This reveals a previously unexplored vulnerability in VLM alignment. To counter this threat, we propose FaithShield Defense, a two-fold framework: (i) a dual-path redundant extension of CLIP that disentangles global and local token contributions, producing explanations more robust to perturbations; and (ii) a novel faithfulness-based detector that verifies explanation reliability via a masking test on top-$k$ salient regions. Explanations that fail this test are flagged as unfaithful. Extensive experiments show that X-Shift reliably compromises explanation faithfulness, while FaithShield restores robustness and enables principled detection of manipulations. Our work formalizes explanation-oriented adversarial attacks and offers a principled defense, enhancing trustworthy and verifiable explainability in VLMs.

## 1 Introduction

Deep Neural Networks (DNNs) play a critical role in modern society, powering applications in healthcare, autonomous vehicles, smart cities, and other safety-critical domains. In particular, Vision–Language Models (VLMs) architectures such as Contrastive Language–Image Pretraining (CLIP) have emerged as foundational models that enable joint reasoning across vision and language (Radford et al., 2021). As these systems are increasingly deployed in high-stakes applications, it is imperative that their predictions are transparent and explainable. Explanation methods, commonly referred to as Explainable AI (XAI), highlight the contribution of input features to model decisions, and are essential for building trust, debugging failures, and identifying spurious correlations (Lipton, 2018; Li et al., 2022; Selvaraju et al., 2017; Li et al., 2025).

Despite their promise, recent studies have demonstrated that explanation methods are themselves vulnerable to manipulation (Kindermans et al., 2019; Ghorbani et al., 2019; Dombrowski et al., 2019; Heo et al., 2019; Slack et al., 2020; Lakkaraju & Bastani, 2020; Huang et al., 2023; Ajalloeian et al., 2023; Kuppa & Le-Khac, 2020). Adversarial perturbations can preserve model predictions while misleading explanations into focusing on irrelevant or incorrect regions. Most prior work has studied this phenomenon in the image domain, targeting gradient-based methods or surrogate explanation models such as LIME and SHAP. However, the vulnerability of XAI in VLMs such as CLIP remains largely unexplored, and no systematic defense mechanisms exist to ensure that explanations are robust or verifiable in this setting (Baniecki & Biecek, 2024). This oversight is critical: in applications like autonomous driving or medical VLMs, explanations directly guide downstream safety logic and human decision-making, so attacks that preserve predictions but shift explanations can meaningfully distort system behavior.

In this work, we address these gaps from two complementary angles. First, we introduce a novel *targeted adversarial attack* on CLIP that manipulates patch–text similarity heatmaps while leaving model results unchanged. Our attack operates in the downstream setting, requiring neither access to training nor modification of evaluation pipelines, thereby closely modeling realistic deployment scenarios.

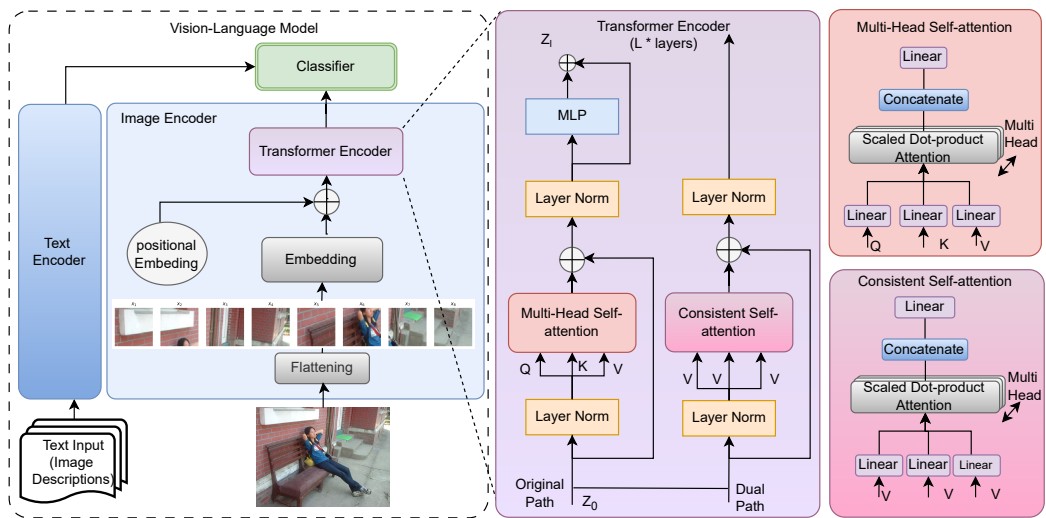

Figure 1: FaithShield Stage I – Dual-path mechanism in the visual transformer, where consistent self-attention operates alongside the standard path to improve heatmap faithfulness and robustness against X-Shift attacks.

Second, we propose a *dual-path redundant extension of CLIP* that disentangles global and local token flows, prunes redundant features, and stabilizes explanation maps against adversarial perturbations. Finally, we integrate a *faithfulness-based detection module* that applies a masking test to identify unfaithful explanation regions by measuring confidence drops, thus enabling a trustworthy and verifiable framework for XAI in VLMs learning.

Our main contributions are as follows:

1. We propose a novel targeted adversarial attack that misleads patch–text heatmaps of CLIP while leaving classification results intact.

2. We design a dual-path redundant extension of CLIP that disentangles feature flows via a self-attention head, removes redundancy, and produces explanations that are robust to adversarial perturbations.

3. We introduce a faithfulness-based detection layer that identifies unfaithful regions in explanation maps, thereby providing a principled mechanism for verifying the trustworthiness of VLMs XAI.

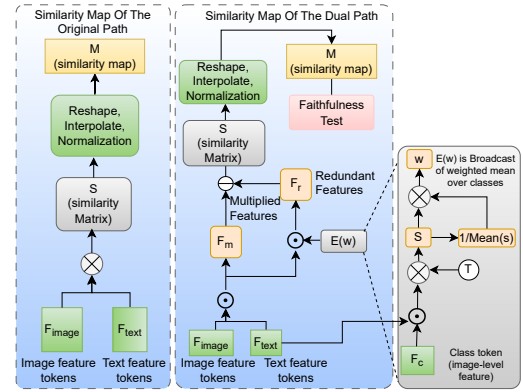

Figure 2: FaithShield workflow with similarity refinement (Stage I) and detection (Stage II).

## 2    RELATED WORK

The susceptibility of deep neural networks to adversarial perturbations is by now well established (Huang et al., 2021; Szegedy et al., 2013; Goodfellow et al., 2014; Carlini & Wagner, 2017; Ilyas

et al., 2018; 2019; Modas et al., 2019; Babadi et al., 2023; Wang et al., 2024; Croce & Hein, 2019; Madry et al., 2017). While the majority of this literature has focused on degrading predictive performance, only recently has research begun to investigate the vulnerability of explanation methods themselves (Baniecki & Biecek, 2024).

Initial studies demonstrated that post hoc explanations are inherently fragile. Kindermans et al. (2019) showed that saliency maps lack invariance to simple input transformations, while Ghorbani et al. (2019) and Dombrowski et al. (2019) revealed that imperceptible perturbations can drastically alter attribution heatmaps without affecting model predictions. Beyond perturbation-based attacks, model-level manipulations have also been explored. For example, Heo et al. (2019) trained networks to mislead attribution methods such as Grad-CAM and LRP, and Slack et al. (2020) demonstrated wrapper-based manipulations of black-box models that arbitrarily control LIME and SHAP explanations, highlighting risks such as *fairwashing* (Lakkaraju & Bastani, 2020).

Building on these findings, subsequent research proposed more targeted attack strategies. Huang et al. (2023) introduced the *Focus-Shifting Attack*, which redirects saliency to adversary-specified regions while preserving prediction consistency. Ajalloeian et al. (2023) developed a sparse perturbation algorithm that manipulates attribution maps more efficiently than $\ell_0$-PGD. In parallel, Kuppa & Le-Khac (2020) studied black-box attacks on LIME and SHAP within cybersecurity applications, establishing an early taxonomy for explanation robustness.

Despite these advances, prior work has largely concentrated on unimodal image classifiers; VLMs remain comparatively underexplored. For CLIP, recent studies have examined adversarial robustness primarily at the level of predictions rather than explanations (Yang et al., 2024). For instance, MP-Nav (Zhang et al.) strengthened poisoning attacks through semantic concept selection, and X-Transfer (Huang et al., 2025b) proposed a universal adversarial perturbation transferable across datasets and tasks. Additional lines of work have addressed backdoor vulnerabilities (Jia et al., 2022), scaling behaviors (Jia et al., 2021), and robustness in grounding tasks (Koh et al., 2023; Huang et al., 2025a).

To the best of our knowledge, no prior work has systematically examined adversarial attacks that specifically manipulate CLIP explanations, nor proposed defenses that simultaneously enhance robustness and detect unfaithful attribution regions. Our work fills this gap by (i) introducing a targeted explanation attack against CLIP and (ii) presenting *FaithShield*, a dual-path framework that disentangles redundant features, improves explanation robustness, and provides a principled detection mechanism for adversarial manipulations.

## 3 X-SHIFT ATTACK OBJECTIVES

We now introduce the **X-Shift attack**, an explanation-focused adversarial strategy that perturbs images such that predictions remain stable while explanation maps are shifted toward a target class. To achieve this, we combine the following complementary objectives: (i) manipulating explanation heatmaps, (ii) preserving the global model output, (iii) enforcing sparsity of perturbations, and (iv) ensuring validity of adversarial examples. Finally, we describe the explainability-focused attack and provide a concrete algorithm.

### 3.1 BASELINE: CLIP MODEL

CLIP (Radford et al., 2021) aligns an image encoder $f_I$ and text encoder $f_T$ in a shared embedding space. Given an image $x$ and text $t$, their normalized embeddings are $z_I = f_I(x)/\|f_I(x)\|_2$, $z_T = f_T(t)/\|f_T(t)\|_2$, with similarity $s(x, t) = z_I^\top z_T$. Training minimizes a symmetric contrastive loss over $N$ image–text pairs:

$$\mathcal{L}_{\text{CLIP}} = \frac{1}{2N} \sum_{i=1}^{N} \left[ -\log \frac{\exp(s(x_i, t_i)/\tau)}{\sum_{j=1}^{N} \exp(s(x_i, t_j)/\tau)} - \log \frac{\exp(s(x_i, t_i)/\tau)}{\sum_{j=1}^{N} \exp(s(x_j, t_i)/\tau)} \right], \quad (1)$$

where $\tau$ is a learnable temperature. Our attack perturbs $x$ into $x_{\text{adv}} = x + \delta$, preserving predictions but shifting explanation maps toward a target class.

## 3.2 ATTACK OBJECTIVES

We combine the following complementary objectives to achieve explanation-focused adversarial perturbations:

**Explanation manipulation.** The primary goal is to force patch embeddings toward the target text embedding. Let $p$ denote the normalized embedding of patch $p$, and $t_{target}$ the target text embedding. Similarity is $s_p = p^\top t_{target}$. We maximize similarity of the top-$K$ patches while suppressing others:

$$\mathcal{L}_{\text{xai}} = -\frac{1}{K} \sum_{i \in \text{TopK}} s_{i,t} + \alpha \cdot \frac{1}{P-K} \sum_{i \notin \text{TopK}} s_{i,t}, \tag{2}$$

where $s_{i,t} = z_i^\top z_{T_{\text{tar}}}$ denotes the similarity between patch embedding $z_i$ and the target text embedding $z_{T_{\text{tar}}}$.

**Prediction preservation.** To prevent label change, we enforce the clean prediction $y^*$ at the global (CLS) level:

$$\mathcal{L}_{\text{pred}} = -\log \frac{\exp(z_{\text{cls}}^\top t_{y^*})}{\sum_c \exp(z_{\text{cls}}^\top t_c)}. \tag{3}$$

**Patch-level margin.** For each patch, the target similarity $s_{p,t}$ must dominate over other classes:

$$\mathcal{L}_{\text{patch}} = \frac{1}{P} \sum_{p=1}^{P} \max\left(0, \max_{c \neq t}(s_{p,c} - s_{p,t} + m)\right), \tag{4}$$

where $s_{p,c} = z_p^\top z_{T_c}$ is the similarity between patch embedding $z_p$ and text embedding $z_{T_c}$.

**Entropy sharpening.** To avoid diffuse attention maps, we encourage sharp similarity distributions:

$$\mathcal{L}_{\text{entropy}} = \sum_{p=1}^{P} m_p \log m_p, \qquad m_p = \frac{\exp(s_{p,t})}{\sum_q \exp(s_{q,t})}, \tag{5}$$

which corresponds to the negative Shannon entropy of the normalized similarities. Minimizing this term encourages sharp and peaked similarity distributions rather than diffuse heatmaps.

**Sparsity constraint.** Perturbations are restricted to $k$ pixels by projecting $\delta = x_{adv} - x$ onto its top-$k$ entries:

$$\delta \leftarrow \text{TopK}(\delta, k). \tag{6}$$

**Validity constraint.** Ensure the adversarial image remains in the valid input domain:

$$x_{adv} \in [0, 1]^d. \tag{7}$$

The total objective combines explanation manipulation with auxiliary constraints:

$$\mathcal{L} = \mathcal{L}_{\text{xai}} + \lambda_{\text{pred}} \mathcal{L}_{\text{pred}} + \lambda_{\text{patch}} \mathcal{L}_{\text{patch}} + \lambda_{\text{ent}} \mathcal{L}_{\text{entropy}} \tag{8}$$

where $\lambda_{\text{pred}}, \lambda_{\text{patch}}$, and $\lambda_{\text{ent}}$ are trade-off coefficients that balance the relative contributions of preserving prediction consistency, enforcing patch-level constraints, and controlling explanation entropy. Tuning these hyperparameters adjusts the strength of each auxiliary objective relative to the main explanation-shifting loss $\mathcal{L}_{\text{xai}}$.

**Explainability Attack Algorithm.** Adversarial examples are generated by iteratively updating the input image using gradient-based optimization. The process is summarized in Algorithm 1 in Appendix A.

## 4 FAITHSHIELD DEFENSE FRAMEWORK

We propose **FaithShield**, a two-stage defense framework designed to counter X-Shift attacks. The framework consists of: (i) a robust explanation module that refines patch embeddings to produce stable heatmaps, and (ii) a faithfulness-based detection mechanism that validates explanation reliability. Together, these components ensure that explanations are both robust and verifiable.

## 4.1 FAITHSHIELD–STAGE I: ROBUST EXPLANATION VIA DUAL-PATH REFINEMENT

Our Stage I design is inspired by the refinement strategies of Li et al. (2025), who introduced consistent attention and redundancy removal to improve the interpretability of CLIP explanations. We adapt these principles but extend them into a *dual-path refinement architecture* that is explicitly tailored to adversarial robustness. Unlike Li et al. (2025), whose focus was interpretability, our formulation integrates three complementary steps: (*i*) consistent self-attention, (*ii*) dual-path feature aggregation, and (*iii*) redundancy elimination, as a unified defense against targeted explanation manipulation.

Let $\{z_p\}_{p=1}^P$ denote the patch embeddings from the vision encoder, and $z_T$ the normalized text embedding. Recall from Section 3.1 that the baseline patch-level similarity is

$$s_p(x,t) = z_p^\top z_T, \quad p = 1, \ldots, P, \tag{9}$$

which can be reshaped into a spatial similarity map. However, such raw maps often highlight background regions (*opposite visualization*) and exhibit class-irrelevant activations (*noisy activations*) across Vision Transformer (ViT) backbones. To mitigate these issues, we build upon the CLIP framework a three-stage refinement procedure: (*i*) consistent self-attention, (*ii*) dual-path feature aggregation, and (*iii*) feature redundancy removal.

**Consistent Self-Attention.** In vanilla CLIP, We follow Li et al. (2025) and replace heterogeneous projections $\phi_q, \phi_k, \phi_v$:

$$A_{\text{raw}} = \sigma(s \cdot QK^\top)V, \quad Q = \phi_q(X), \ K = \phi_k(X), \ V = \phi_v(X), \tag{10}$$

which may relate tokens from semantically inconsistent regions. We instead employ a homogeneous projection $\phi_v$ to enforce semantic consistency:

$$A_{\text{con}} = \sigma(s \cdot VV^\top)V, \quad V = \phi_v(X). \tag{11}$$

This ensures that self-attention emphasizes tokens with coherent semantics, verified quantitatively via the mean Foreground Selection Ratio (mFSR). Figure 1 illustrates the dual-path schema, highlighting the replacement of raw multi-head self-attention with consistent self-attention blocks to ensure more coherent token interactions.

**Dual-Path Refinement.** Not all intermediate modules are equally aligned with the final prediction. Affinity between text features $F_t$ and block-level class token features $\hat{F}_c$ is measured as

$$a(F_t, \hat{F}_c) = \frac{1}{N_t} \sum_{i=1}^{N_t} F_t^{(i)} \hat{F}_c, \tag{12}$$

revealing that feed-forward networks (FFNs) often drift toward negatives and harm interpretability. We therefore aggregate only consistent self-attention modules, skipping FFNs via a dual-path architecture:

$$\hat{x}_{i+1} = \begin{cases} \text{None}, & i < d, \\ f_{A_{\text{con}}}(x_i, \phi_v) + x_i, & i = d, \\ f_{A_{\text{con}}}(x_i, \phi_v) + \hat{x}_i, & i > d, \end{cases} \tag{13}$$

while preserving the original path $x_{i+1}$ for final model outputs. This design enhances interpretability without degrading recognition accuracy (Li et al., 2025).

**Feature Redundancy Removal** Noisy activations arise from redundant features shared across categories. Based on (Li et al., 2025), we first compute multiplied features:

$$F_m = \mathcal{E}(F_i) \odot \mathcal{E}(F_t), \quad F_m \in \mathbb{R}^{N_i \times N_t \times C}, \tag{14}$$

where $F_i$ and $F_t$ are L2-normalized image and text features, $\odot$ denotes element-wise product, and $\mathcal{E}$ broadcasts to matching shape. Next, we reweight influential classes:

$$s = \sigma(\tau \cdot F_c F_t^\top), \quad w = \frac{s}{\mu_s}, \tag{15}$$

where $F_c$ is the class token, $\tau$ is a logit scale, and $\mu_s$ the mean of $s$. Redundant features are then estimated as

$$F_r = \text{mean}(F_m \odot \mathcal{E}(w)) \in \mathbb{R}^{N_i \times C}, \tag{16}$$

and subtracted:

$$S = \text{sum}(F_m - \mathcal{E}(F_r)) \in \mathbb{R}^{N_i \times N_t}. \tag{17}$$

Finally, $S$ is reshaped, interpolated, and normalized to produce the refined similarity map.

**Final Heatmap.** The refined patch–text similarity is normalized via softmax:

$$M(x,t)[p] = \frac{\exp(\alpha\, s_p^{\text{ref}}(x,t))}{\sum_{q=1}^{P} \exp(\alpha\, s_q^{\text{ref}}(x,t))}, \tag{18}$$

where $\alpha$ controls sharpness. This yields heatmaps that are semantically faithful, less noisy, and more foreground-focused. Algorithm 2 in Appendix B illustrates the workflow of this subsection.

## 4.2 FAITHSHIELD–STAGE II: FAITHFULNESS-BASED DETECTION

The second stage of FaithShield introduces a novel detection module that tests whether an explanation is truly faithful to the model's decision. While prior work has focused on refining attention maps to improve interpretability, none has provided a systematic mechanism for *detecting adversarially misleading explanations*. Our Stage II addresses this gap.

Even with refined embeddings, adversarial perturbations may still redirect saliency toward irrelevant regions while leaving the prediction intact. To flag such cases, we propose a *confidence-drop test*: mask the top-$k$ most salient regions indicated by the explanation and re-evaluate the model's confidence for the target class. For a faithful explanation, removing the highlighted regions should cause a substantial confidence drop, reflecting causal alignment between the explanation and the prediction. Conversely, if the confidence remains nearly unchanged, the heatmap is identified as misleading.

Given a heatmap $M(x,t)$ for class $t$, we select the top-$\rho\%$ patches:

$$\mathcal{M}_t = \{p \mid M(x,t)[p] \geq \tau_t\}, \tag{19}$$

where $\tau_t$ is chosen such that $|\mathcal{M}_t| = \rho \cdot P$. These patches are suppressed in the input image to form a perturbed version $x'$:

$$x' = \begin{cases} x \odot (1 - M_t), & \text{(zeroing)} \\ \text{Blur}(x \odot M_t) + x \odot (1 - M_t), & \text{(blurring),} \end{cases} \tag{20}$$

where $M_t$ is upsampled to image resolution.

We then measure cosine similarity before and after masking:

$$s_{\text{orig}} = z_I^\top z_T, \qquad s_{\text{masked}} = (z_I')^\top z_T, \tag{21}$$

where $z_I = f_I(x)/\|f_I(x)\|$ and $z_I' = f_I(x')/\|f_I(x')\|$. Since $s(x,t)$ is a cosine similarity in $[-1, 1]$, we normalize it into $[0, 1]$ for interpretability when measuring confidence:

$$\text{conf}(s) = \tfrac{1}{2}(1 + s). \tag{22}$$

This normalization does not affect the ranking of similarities but enables a consistent interpretation of $\Delta_{\text{conf}}$ as a probability drop. the confidence drop is defined as:

$$\Delta_{\text{conf}} = \text{conf}(s_{\text{orig}}) - \text{conf}(s_{\text{masked}}). \tag{23}$$

If the masked region is truly explanatory, $\Delta_{\text{conf}}$ will be large. Conversely, if $\Delta_{\text{conf}}$ is small, the explanation is deemed unfaithful. We flag misleading explanations whenever:

$$\Delta_{\text{conf}} < \theta, \tag{24}$$

with threshold $\theta$. The overall defense integrates two complementary modules:

1. **Robust explanation:** Dual-path refinement of patch embeddings yields faithful and stable similarity maps.
2. **Faithfulness detection:** Masking-based tests on clean and adversarial images identify unfaithful regions.

Together, these modules ensure that explanations are both *robust* and *verifiable*. The procedure is summarized in Algorithm 3 in Appendix B. Figure 2 illustrates the refinement of similarity maps through dual-path processing and feature redundancy removal, followed by the application of faithfulness-based detection.

## 5 EXPERIMENTS

Our evaluation is designed to answer the following research questions:

- How effective is the proposed attack in shifting XAI?
- Does the dual-path refinement improve robustness of XAI under adversarial perturbations?
- Can the faithfulness-based detection reliably identify misleading XAI?

**Models and Datasets.** We evaluate our attack and defense framework at inference time, without requiring additional training data. Experiments are conducted on the validation splits of three benchmark datasets: ImageNet-1k (Deng et al., 2009), Flickr30k (Young et al., 2014), and MS-COCO (Chen et al., 2015), which provide diverse natural images and object-level annotations for assessing VLMs explanations. For models, we utilize the CLIP family of vision–language encoders, specifically ViT-B/16 (Radford et al., 2021), ViT-B/32 (Radford et al., 2021), and ViT-L/14 (Dosovitskiy et al., 2020), which span a range of capacities and input resolutions to assess the generality of our attack and defense across different backbones.

**Implementation.** We implement attack and defense on official CLIP models, using patch–text similarity maps that compute cosine similarity between patch and text embeddings. Unlike gradient-based attributions (e.g., Grad-CAM, Integrated Gradients), which often yield unstable ViT heatmaps, similarity maps are faithful, text-conditioned, efficient (single forward pass), and deterministic. CLIP employs attention pooling, yielding a $7 \times 7$ grid for $224 \times 224$ inputs (datasets resized accordingly). The attack loss follows Section 3, with weights 20.0 for $\mathcal{L}_{\mathrm{xai}}$, $\lambda_{\mathrm{ent}}$ for entropy, $\lambda_{\mathrm{margin}}$ for patch separation, and $0.01\lambda_{\mathrm{pred}}$ for prediction consistency, tuned to balance manipulation and stability.

**Metrics.** We evaluate global prediction stability and explanation robustness using four quantitative metrics: CosSim (CLS), Max $\Delta$Prob, and IoU (Top-$k$). Formal definitions of these metrics are provided in Appendix C.1.

### 5.1 RESULTS ON EXPLAINABILITY

**Proposed Attack Effectiveness.** Figure 3 demonstrates that the X-Shift adversarial perturbations successfully shift CLIP's explanation maps while preserving the predicted label. In the clean case, the heatmap correctly attends to the input concept (e.g., "bench"), whereas under the X-Shift attack the attention is redirected toward unrelated regions (e.g., the "wall"), thereby compromising explanation faithfulness. Stage I of the FaithShield defense is also shown, illustrating improved robustness of the heatmaps under adversarial perturbations.

Furthermore, Figures 4, 5, and 6 visualize additional examples from ImageNet, Flickr30k, and COCO. In each case, the perturbation remains imperceptible to humans yet induces substantial shifts in the explanation maps, highlighting the vulnerability of current XAI methods.

**Robustness and Detection with FaithShield.** Figures 4, 5, and 6 further demonstrate the effectiveness of the FaithShield framework. Stage I consistently improves robustness by preserving faithful heatmaps even under adversarial perturbations. In addition, the faithfulness-based detection module successfully flags regions that are inconsistent with the input text, identifying adversarially induced shifts toward unrelated areas. These results confirm that FaithShield not only mitigates explanation manipulation but also provides a reliable mechanism to detect when explanations have been compromised.

Figure 3: Visualization of a sample image under the X-Shift attack and FaithShield. Columns show the clean and adversarial images (optimized to shift CLIP's explanation toward "ground" while keeping the "bench" prediction), the clean–adversarial difference map, CLIP heatmaps showing explanation drift, and FaithShield Stages I–II, which suppress the drift and reveal unrelated manipulated regions.

**Quantitative Evaluation.** Table 1 summarizes results across ImageNet, Flickr30k, and MS-COCO with three CLIP backbones (ViT-B/16, ViT-B/32, ViT-L/14). Across all settings, the **CosSim (CLS)** remains high (typically $\geq 0.93$) and the **Max $\Delta$Prob** is nearly zero, confirming that the X-Shift perturbations preserve the global classification decision. The main differences arise in explanation stability. For vanilla CLIP, the **Top-$k$ IoU** between clean and adversarial heatmaps is consistently low (e.g., $0.487$ on ImageNet ViT-B/16, $0.727$ on Flickr30k ViT-L/14, and $0.556$ on COCO ViT-B/32), revealing that explanations are highly sensitive to perturbations even when predictions remain unchanged. By contrast, **FaithShield** substantially improves alignment between clean and adversarial maps, achieving IoU gains of $+0.124$ (ImageNet ViT-B/16), $+0.222$ (Flickr30k ViT-L/14), and $+0.346$ (COCO ViT-B/16). These improvements consistently hold across datasets and backbones, with relative gains often exceeding 20–35%. Taken together, the results demonstrate that FaithShield effectively mitigates explanation shifts induced by adversarial perturbations, delivering robust and reliable XAI without compromising classification accuracy.

**Evaluation of FaithShield Ablations.** Our empirical findings align with the architectural ablations reported in Appendix C.5. Individually, the Stage-I components (S1, S2, FS) offer only partial stability, producing IoU values in the range of $0.70$–$0.88$. In contrast, the full Stage-I + Stage-II pipeline achieves substantially stronger and prediction-preserving alignment between clean and adversarial explanations, with IoU improving to $0.90$–$0.97$. These results confirm that robust explanation consistency emerges *only* when structural refinement (Stage I) is paired with the $\Delta$conf-based causal detector (Stage II).

**Evaluation of X-Shift Attack Transferability.** Our cross-model analysis (Appendix C.2) demonstrates that X-Shift perturbations generalize across CLIP backbones and explainability methods. Self-attacks produce the strongest manipulation ($\text{IoU}_{\text{TopK}}$ as low as $0.44$–$0.47$), while cross-model transfer remains strong. For instance, a perturbation crafted on ViT-B/32 transfers to ViT-L/14 with IoU $= 0.63$. Additionally, ScoreCAM, RISE, and gradient-based attribution maps all exhibit consistent explanation drift under X-Shift, indicating that the attack corrupts the shared image–text embedding space, not a specific explainer. When FaithShield is applied, these drifts are dramatically reduced across all architectures and XAI methods (Appendix C.3).

**Evaluation of Attack-Loss Ablations.** Appendix C.4 analyzes the effect of removing each loss component in the X-Shift objective. The patterns are consistent across datasets:

- Removing the XAI-shift term weakens the attack, increasing IoU (e.g., $0.78 \rightarrow 0.72$) and reducing TargetSim.

- Removing prediction-stability terms (e.g., $\mathcal{L}_{\text{pred}}$) breaks stealth, increasing Max$\Delta$Prob by nearly an order of magnitude (from $6.6 \times 10^{-5}$ to $5.4 \times 10^{-4}$).

- Using only the XAI term yields the strongest drift (IoU $\approx 0.82$) but destroys classification stability.

- The full objective achieves the best balance: strong manipulation (IoU $\approx 0.79$), high TargetSim, stable CLS embedding ($0.977$), and minimal Max$\Delta$Prob.

Together, these results show that **FaithShield counters both direct and transferable explanation attacks**, and that **each component of the X-Shift loss and each stage of FaithShield are necessary and complementary**. The system delivers robust, prediction-preserving interpretability across datasets, architectures, and XAI techniques.

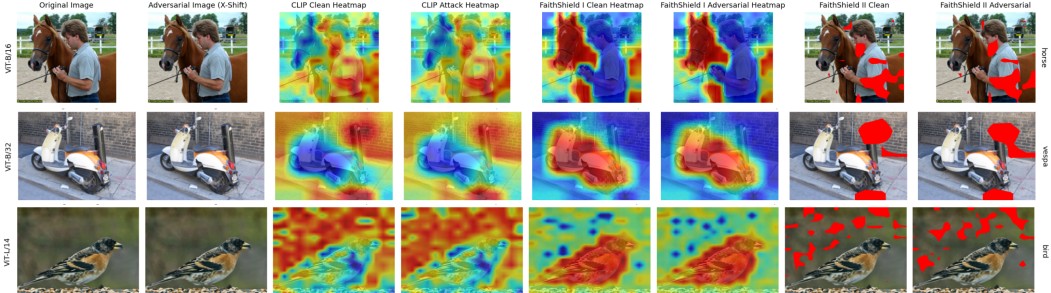

Figure 4: Comparison of CLIP explanations on ImageNet dataset(ViT-B/16, ViT-B/32, ViT-L/14) under X-Shift attack and FaithShield defense. Columns show original/adversarial images, CLIP heatmaps, and FaithShield stages I and II (clean vs. adversarial).

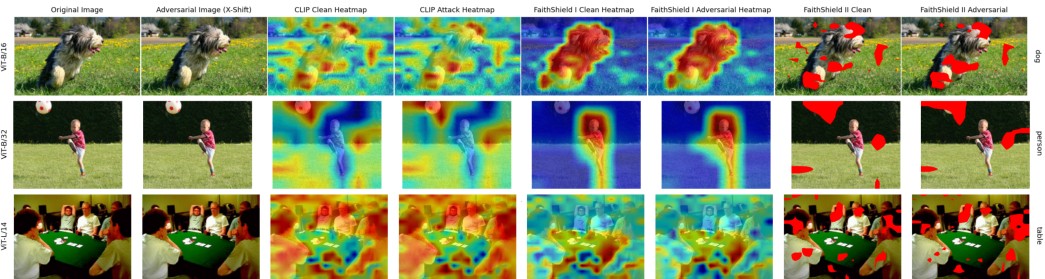

Figure 5: Explanations on Flickr30k samples using CLIP (ViT-B/16, ViT-B/32, ViT-L/14) under X-Shift attack and FaithShield defense. Shown are original/adversarial images, CLIP heatmaps, and FaithShield stages I and II (clean vs. adversarial).

Table 1: Quantitative comparison of **Vanilla CLIP** vs. **FaithShield** under X-Shift attack across datasets and backbones. Metrics: cosine similarity (CosSim), maximum probability change under X-Shift attack (Max $\Delta$Prob), and Top-$k$ IoU.

| Dataset | Backbone | Vanilla CLIP | | | FaithShield | | |
|---------|----------|--------|------------|-----|--------|------------|-----|
| | | CosSim | Max $\Delta$Prob | IoU | CosSim | Max $\Delta$Prob | IoU |
| ImageNet | ViT-B/16 | 0.805 | 0.004 | 0.487 | **0.805** | **0.004** | **0.611** |
| | ViT-B/32 | 0.807 | 0.004 | 0.450 | **0.807** | **0.004** | **0.634** |
| | ViT-L/14 | 0.948 | 0.000 | 0.551 | **0.948** | **0.000** | **0.877** |
| Flickr30k | ViT-B/16 | 0.935 | 0.000 | 0.841 | **0.935** | **0.000** | **0.933** |
| | ViT-B/32 | 0.974 | 0.000 | 0.867 | **0.974** | **0.000** | **1.000** |
| | ViT-L/14 | 0.933 | 0.000 | 0.727 | **0.933** | **0.000** | **0.949** |
| MS-COCO | ViT-B/16 | 0.977 | 0.000 | 0.611 | **0.977** | **0.000** | **0.902** |
| | ViT-B/32 | 0.953 | 0.000 | 0.556 | **0.953** | **0.000** | **0.867** |
| | ViT-L/14 | 0.962 | 0.000 | 0.583 | **0.962** | **0.000** | **0.727** |

## 6 CONCLUSION

This paper examined the vulnerability of VLMs, focusing on CLIP, to adversarial explanation attacks. We introduced X-Shift, a targeted perturbation that manipulates patch–text heatmaps without altering classification outputs, exposing a fundamental weakness of current explanation mechanisms: explanations can be redirected toward irrelevant regions while predictions remain unchanged. To address this, we proposed *FaithShield*, a dual-path refinement combined with a faithfulness-based detection module. The refinement stabilizes explanation maps by disentangling redundant feature flows, while the detection mechanism applies a causal masking test to flag unfaithful regions. Together, they provide robust and verifiable explanations under adversarial perturbations. Our findings highlight the need for trustworthy and accountable VLMs. Future work will extend this framework

Figure 6: Explanation robustness on COCO samples using CLIP (ViT-B/16, ViT-B/32, ViT-L/14) under X-Shift attack and FaithShield defenses. Columns display original vs. adversarial images, CLIP heatmaps, and FaithShield stages I & II (clean vs. adversarial).

to other foundation models, evaluate resilience against adaptive attacks, and explore applications in safety-critical domains such as autonomous driving and medical decision support.

## REPRODUCIBILITY STATEMENT

All implementation details, including training and evaluation scripts, are provided in the anonymized supplementary file (`supplementary_code.zip`). This ensures reproducibility while maintaining anonymity during the review process.

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

# A   THE X-SHIFT ATTACK ALGORITHM

The X-Shift attack (Algorithm 1) implements the objectives defined in Section 3, perturbing inputs to shift explanation maps while preserving the original prediction.

---

**Algorithm 1** X-Shift Attack: Explanation Manipulation on CLIP

---

**Input:** clean image $x$, text embeddings $\{t_c\}$, target index $t$, step size $\eta$, sparsity $k$, iterations $T$
**Output:** adversarial image $x^{adv}$
Initialize $x^{(0)} \leftarrow x$
**for** $i = 1$ to $T$ **do**
    Compute patch embeddings $\{z_p\}$ and CLS embedding $z_{cls}$
    Evaluate losses $\mathcal{L}_{xai}, \mathcal{L}_{pred}, \mathcal{L}_{patch}, \mathcal{L}_{entropy}$
    Total loss:
$$\mathcal{L} \leftarrow \mathcal{L}_{xai} + \lambda_{pred}\mathcal{L}_{pred} + \lambda_{patch}\mathcal{L}_{patch} + \lambda_{ent}\mathcal{L}_{\text{entropy}}$$
    Gradient update:
$$x^{(i)} \leftarrow x^{(i-1)} - \eta \cdot \text{sign}(\nabla_x \mathcal{L})$$
    Sparsity projection:
$$\delta \leftarrow \text{TopK}(x^{(i)} - x^{(0)}, k), \quad x^{(i)} \leftarrow x^{(0)} + \delta$$
    Clamp to valid domain:
$$x^{(i)} \leftarrow \text{clip}(x^{(i)}, 0, 1)$$
**end for**
**return** $x^{adv} = x^{(T)}$

---

# B   THE FAITHSHIELD ALGORITHMS

FaithShield Stage I (Algorithm 2) refines explanation heatmaps using consistent self-attention, dual-path aggregation, and feature redundancy removal, as described in Section 4.1.

---

**Algorithm 2** FaithShield – Stage I: Dual-Path Refinement for Robust Explanations

---

**Input:** $x$ (image), $t$ (text), $f_I$ (vision encoder), $f_T$ (text encoder), $d$ (depth), $\alpha$ (temperature)
**Output:** Refined explanation heatmap $M(x, t)$
**Step 1: Encode.** Extract patch features $F_i = f_I(x)$ and text features $F_t = f_T(t)$.
**Step 2: Consistent attention.** Replace raw attention with consistent self-attention:
$$A_{\text{con}} = \sigma(sVV^\top)V$$
**Step 3: Dual path aggregation.** From depth $d$, aggregate consistent attention outputs:
$$\hat{x}_{i+1} = f_{A_{\text{con}}}(x_i, \phi_v) + \hat{x}_i$$
**Step 4: Feature redundancy removal.** Fuse image and text features:
$$F_m = \mathcal{E}(F_i) \odot \mathcal{E}(F_t)$$
Remove redundant features $F_r$ (see Eq. (10)), yielding:
$$S = \text{sum}(F_m - \mathcal{E}(F_r))$$
**Step 5: Heatmap.** Normalize $S$ and apply softmax with $\alpha$ to obtain $M(x, t)$.
**return** $M(x, t)$

---

FaithShield Stage II formalizes the confidence-drop test in algorithmic form, based on the mathematical definitions in Section 4.2.

---

**Algorithm 3** FaithShield – Stage II: Faithfulness-Based Detection (mathematical form)

---

**Input:** image $x$, adversarial image $x^{adv}$, text embeddings $\{z_{T_j}\}_{j=1}^N$, threshold $\theta$, masking ratio $\rho$
**Output:** misleading explanation flags per label
**for** $j = 1$ to $N$ **do**
    Compute heatmap $M(x, t_j)$
    Select top-$\rho$% patches:
$$\mathcal{M}_j = \{\, p \mid M(x, t_j)[p] \geq \tau_j \,\}, \quad |\mathcal{M}_j| = \rho P$$
    Mask regions to obtain perturbed input:
$$x'_j = x \odot (1 - M_j) \quad \text{or} \quad x'_j = \text{Blur}(x \odot M_j) + x \odot (1 - M_j)$$
    Compute similarities:
$$s_j^{orig} = z_I^\top z_{T_j}, \quad s_j^{masked} = (z'_I)^\top z_{T_j}$$
with $z_I = f_I(x)/\|f_I(x)\|$, $z'_I = f_I(x'_j)/\|f_I(x'_j)\|$
    Normalize to confidence:
$$\text{conf}(s) = \tfrac{1}{2}(1 + s)$$
    Compute confidence drop:
$$\Delta_j^{conf} = \text{conf}(s_j^{orig}) - \text{conf}(s_j^{masked})$$
    Flag $t_j$ as misleading if:
$$\Delta_j^{conf} < \theta$$
**end for**
**return** flags for all labels $t_j$

---

## C   Extended Experimental Analysis

This appendix provides the complete definitions of all quantitative metrics used in Section 5, followed by expanded experimental results that analyze cross-architecture transferability, ablation studies, FaithShield component isolation, and adaptive-attacker robustness.

### C.1   Evaluation Metrics

We measure four complementary aspects of model behavior under X-Shift perturbations: (i) embedding-level stealth, (ii) classifier stability, (iii) spatial attribution consistency at the patch level, and (iv) distributional similarity of the full explanation map. Below we summarize the exact formulations.

**Cosine Similarity of CLS Tokens (CosSim $\uparrow$).**   This metric quantifies how close the clean and adversarial global embeddings remain. A high value indicates a *stealthy* attack that preserves high-level semantics. Given the CLS embeddings $z_{\text{clean}}$ and $z_{\text{adv}}$:

$$\text{CosSim}_{\text{CLS}} = \frac{z_{\text{clean}} \cdot z_{\text{adv}}}{\|z_{\text{clean}}\|_2 \, \|z_{\text{adv}}\|_2}. \tag{25}$$

**Maximum Probability Deviation (Max$\Delta$Prob $\downarrow$).**   This term measures the largest change in predicted probability across all text prompts. Low values imply that classification remains unchanged even though the explanation map shifts:

$$\text{Max}\,\Delta\text{Prob} = \max_j |P(y_j \mid x_{\text{clean}}) - P(y_j \mid x_{\text{adv}})|. \tag{26}$$

**Intersection-over-Union of Top-$k$ Patches (IoU-Top$k$ $\downarrow$).**   We extract the top-$k$ highest-scoring patches in the similarity map for a target concept, compute the corresponding binary masks $M_{\text{clean}}$

and $M_{\text{adv}}$, and evaluate. Lower IoU indicates *stronger spatial manipulation*, as fewer top patches are preserved under the adversarial perturbation. We use either a fixed $k$ or a percentage $k = \alpha HW$ of all patches:

$$\text{IoU}_{\text{Top-}k} = \frac{|M_{\text{clean}} \cap M_{\text{adv}}|}{|M_{\text{clean}} \cup M_{\text{adv}}|}. \tag{27}$$

**Soft Intersection-over-Union (Soft-IoU $\downarrow$).**    To capture distributional differences beyond hard top-$k$ sets, we compute a soft approximation using a temperature $\tau$. This measures global distributional drift, complementing IoU-Top$k$:

$$p_{\text{clean}} = \text{softmax}(s_{\text{clean}}/\tau), \qquad p_{\text{adv}} = \text{softmax}(s_{\text{adv}}/\tau), \tag{28}$$

$$\text{Soft-IoU} = \frac{\sum_i \min(p_{\text{clean},i}, p_{\text{adv},i})}{\sum_i \max(p_{\text{clean},i}, p_{\text{adv},i})}. \tag{29}$$

**Spearman Rank Correlation (Spearman).**    We compute the rank correlation between the flattened similarity maps. Low correlation indicates large reordering of influential patches:

$$\rho = \text{Spearman}(s_{\text{clean}}, s_{\text{adv}}). \tag{30}$$

**Wasserstein Distance (EMD).**    We compute the Earth Mover's Distance between flattened similarity scores. EMD captures how much "work" is needed to transform the clean explanation distribution into its adversarial counterpart:

$$\text{EMD}(s_{\text{clean}}, s_{\text{adv}}) = W_1(s_{\text{clean}}, s_{\text{adv}}). \tag{31}$$

Together, these metrics provide a multi-dimensional characterization of explanation-shifting behavior: *stealth* (CosSim, Max$\Delta$Prob), *local spatial reordering* (IoU-Top$k$), and *global distributional drift* (Soft-IoU, Spearman, EMD).

## C.2    TRANSFERABILITY OF X-SHIFT ACROSS VISION TRANSFORMER BACKBONES

We evaluate whether explanation-shifting perturbations generated on one CLIP encoder transfer to other CLIP variants with different patch sizes and embedding dimensions. Specifically, we test ViT-B/16, ViT-B/32, and ViT-L/14 models in a source-to-target setting, measuring:

- Cosine similarity between clean and adversarial CLS embeddings ($\text{CosSim}_{\text{CLS}}$)
- Maximum deviation in predicted probabilities across all text prompts ($\text{Max}\Delta\text{Prob}$)
- Patch-level shift in the similarity map for the target concept using $\text{IoU}_{\text{Top-}k}$ (lower is better for measuring explanation manipulation)
- Smooth distributional similarity shift using Soft-IoU (also lower is better)

**Experiment analysis.**    Table 2 shows that self-attacks produce the lowest IoU-TopK values (0.44–0.47), indicating strong spatial manipulation of the similarity map without altering model predictions (CosSim $>0.94$, Max$\Delta$Prob$< 4 \times 10^{-4}$). Cross-architecture transfer is moderate but consistent: for example, perturbations crafted on ViT-B/32 transfer to ViT-L/14 with IoU$= 0.63$, demonstrating that the attack generalizes across patch sizes (14–32) and embedding widths. Soft-IoU remains high because CLIP map distributions are smooth, but localized top-$k$ patch ordering is reliably perturbed. Overall, the results confirm that X-Shift attacks preserve classification while inducing model-invariant explanation shifts. Figure 7 visualizes the mean IoU-TopK transfer matrix (lower is better), highlighting asymmetric transfer patterns: perturbations from ViT-B/32 transfer more strongly to other backbones than those from ViT-L/14. The heatmap corroborates the numerical

Table 2: Transferability of explanation-shifting perturbations across CLIP architectures. We report cosine similarity of CLS tokens (CosSim↑), maximum change in predicted probability (Max$\Delta$Prob↓), and patch-overlap metrics IoU-TopK and Soft-IoU (both "lower is better" for capturing successful heatmap manipulation). Self-attacks achieve the lowest IoU (largest shift), while cross-model transfer remains moderate but consistent across backbones.

| Source | Target | CosSim↑ | Max$\Delta$Prob↓ | IoU-TopK↓ | Soft-IoU↓ | Spearman | EMD |
|---|---|---|---|---|---|---|---|
| | ViT-L/14 | 0.9421 | 0.00044 | **0.4713** | 0.9837 | 0.7710 | 0.0062 |
| ViT-L/14 | ViT-B/16 | **0.9928** | **0.00007** | 0.7818 | 0.9962 | 0.9496 | 0.0010 |
| | ViT-B/32 | 0.9180 | 0.00023 | 0.8571 | **0.9973** | **0.9914** | 0.0017 |
| | ViT-L/14 | 0.9805 | 0.00013 | 0.6842 | 0.9915 | 0.8940 | 0.0024 |
| ViT-B/16 | ViT-B/16 | 0.7628 | 0.00039 | **0.4412** | 0.9891 | 0.7755 | 0.0104 |
| | ViT-B/32 | 0.9721 | 0.00029 | 0.7059 | 0.9907 | 0.9194 | 0.0032 |
| | ViT-L/14 | 0.9520 | 0.00017 | 0.6316 | 0.9910 | 0.8743 | 0.0175 |
| ViT-B/32 | ViT-B/16 | 0.9933 | 0.00026 | **0.5882** | 0.9902 | 0.9283 | 0.0060 |
| | ViT-B/32 | 0.9278 | **0.00009** | 0.5750 | 0.9896 | 0.8442 | 0.0128 |

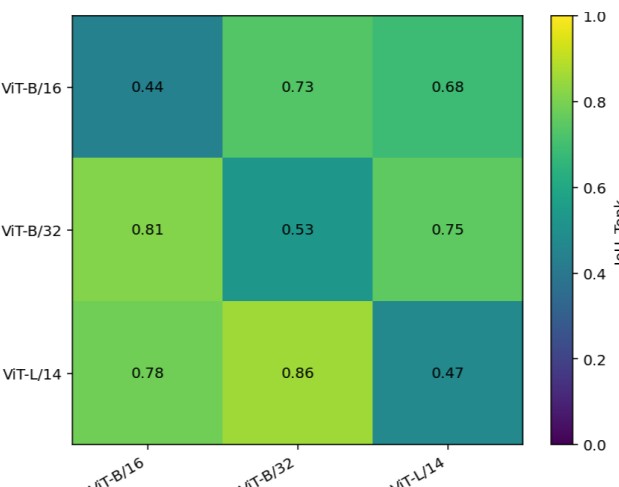

Figure 7: Transfer matrix of IoU-TopK (lower indicates stronger manipulation) across source → target CLIP backbones. ViT-B/32 perturbations transfer most broadly, while ViT-L/14 perturbations remain more model-specific.

results in Table 2 and illustrates which source architectures most reliably induce cross-model explanation shifts.

## C.3 TRANSFERABILITY ACROSS EXPLAINABILITY METHODS

To evaluate whether targeted explainability attacks generated on vanilla CLIP transfer across attribution methods, datasets, and architectures, we compute a grid of heatmaps using our visualization pipeline. The adversary optimizes the X-Shift perturbation directly on CLIP ViT-B/16, relocating the patch–text similarity mass from the clean concept (e.g., "cat") toward an adversarial concept (e.g., "background") while preserving the original prediction. Thus, any changes observed in Figures 8–10 reflect explanation drift rather than classification errors.

**Choice of XAI methods.** We include ScoreCAM, RISE, and gradient-based explanation (GAE) *solely* to assess attack transferability. These attribution methods were originally designed for single-stream CNN classifiers and do not model CLIP's multimodal text-image alignment or transformer attention. Consequently, they tend to produce diffuse and low-fidelity maps on both clean and adversarial CLIP inputs. Their purpose here is diagnostic: to show that VLMs such as CLIP require

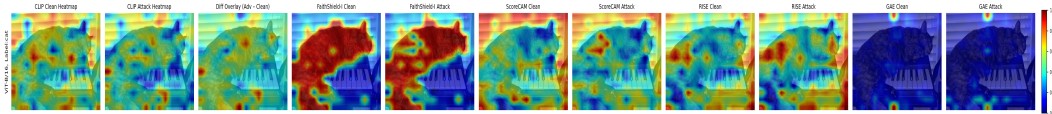

Figure 8: XAI transferability results for the X-Shift adversarial perturbation on COCO dataset with CLIP ViT-B/16. Similarity maps, ScoreCAM, RISE, and GAE all exhibit explanation drift under the attack when applied to vanilla CLIP. FaithShield Stage I, however, suppresses this drift and produces nearly identical clean and adversarial heatmaps, confirming its robustness to explanation-level manipulation.

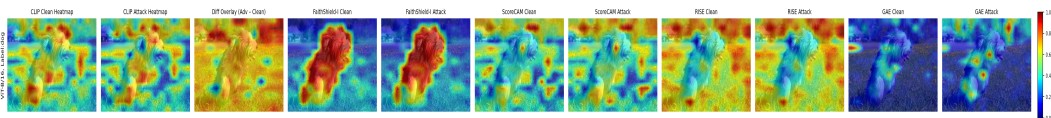

Figure 9: XAI transferability results for the X-Shift adversarial perturbation on the Flickr30k dataset with CLIP ViT-B/16. Similarity maps, ScoreCAM, RISE, and GAE all exhibit explanation drift under the attack when applied to vanilla CLIP. FaithShield Stage I, however, suppresses this drift and produces nearly identical clean and adversarial heatmaps, confirming its robustness to explanation-level manipulation.

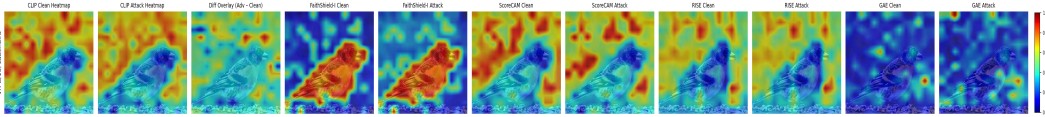

Figure 10: XAI transferability results for the X-Shift adversarial perturbation on the ImageNet dataset with CLIP ViT-B/16. Similarity maps, ScoreCAM, RISE, and GAE all exhibit explanation drift under the attack when applied to vanilla CLIP. FaithShield Stage I, however, suppresses this drift and produces nearly identical clean and adversarial heatmaps, confirming its robustness to explanation-level manipulation.

dedicated, reliable, and modality-aware explanation tools, and that CNN-based attribution methods lack the grounding needed to produce trustworthy heatmaps for multimodal models.

We evaluate this effect across COCO (Figure 8), Flickr30K (Figure 9), and ImageNet (Figure 10), plotting similarity maps and their differences for both clean and adversarial images. In addition to CLIP's native patch–text similarity heatmaps, we generate ScoreCAM, RISE, and Gradient-based Explanation (GAE) maps to test cross-method transferability. For vanilla CLIP, the X-Shift perturbation consistently alters the spatial attribution structure: the clean similarity map highlights the true object regions, whereas the adversarial map redirects attention toward background patches aligned with the attacker's target text. This drift appears across all datasets and attribution methods, and the difference-overlay visualizations clearly reveal large, structured regions of displaced saliency.

In contrast, FaithShield Stage-I demonstrates strong resistance to the X-Shift attack. Across all datasets, its clean and adversarial similarity maps remain visually aligned, and the overlays exhibit only sparse, low-intensity deviations. Furthermore, ScoreCAM, RISE, and GAE generated on top of the FaithShield encoder show similarly stable behavior, indicating that the robustness achieved by Stage-I transfers to downstream explainability tools as well. This confirms that FaithShield's consistent self-attention and redundancy-suppression mechanisms effectively block adversarial similarity-map manipulation, preventing the attack from propagating across XAI methods and across datasets.

## C.4 ABLATION OF THE X-SHIFT ATTACK OBJECTIVE

To validate that each component of the adversarial objective in Section 3 (Equation 8) is necessary for constructing a stable and optimized X-Shift attack, we perform a controlled ablation over the

four loss terms:

$$\mathcal{L}_{\mathrm{xshift}} = \mathcal{L}_{\mathrm{xai}} + \lambda_{\mathrm{pred}}\mathcal{L}_{\mathrm{pred}} + \lambda_{\mathrm{ent}}\mathcal{L}_{\mathrm{ent}} + \lambda_{\mathrm{margin}}\mathcal{L}_{\mathrm{margin}}. \qquad (32)$$

As shown in Table 3, each ablation variant optimizes the same adversarial direction toward the specified target text embedding, but differs in which loss components from Eq. (8) are enabled. We report four complementary metrics for evaluating the resulting adversarial examples: (i) CLS embedding deviation, measured by $\mathrm{CosSim}_{\mathrm{CLS}}$; (ii) maximum change in class probabilities ($\mathrm{Max}\Delta\mathrm{Prob}$), which captures prediction stability; (iii) patch-level drift, quantified as the IoU of the top-$k$ most salient patches ($\mathrm{IoU}_{\mathrm{Topk}}$) between clean and adversarial similarity maps; and (iv) final similarity to the adversarial target text (TargetSim), which reflects the strength of the explanation manipulation. Together, these metrics allow us to isolate the contribution of each loss term and assess the necessity of all components of the X-Shift objective. Figure 11 illustrates how each loss component in the X-Shift objective contributes to explanation manipulation: the full loss achieves controlled but meaningful heatmap displacement, the xai-only loss exaggerates the shift at the cost of prediction stability, and the pred-only loss barely alters the explanation, confirming that the explanation-shift terms are necessary for targeted manipulation.

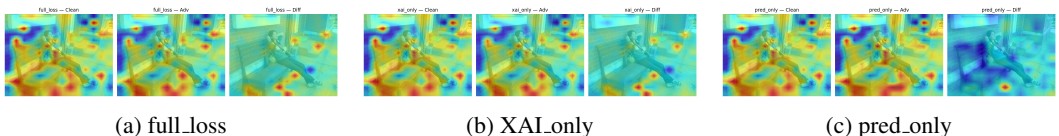

(a) full_loss            (b) XAI_only            (c) pred_only

Figure 11: Ablation of the three X-Shift loss components. *full_loss* produces a balanced, targeted explanation shift; *xai_only* yields strong drift but breaks prediction consistency; *pred_only* preserves the clean map with minimal drift.

Table 3: Ablation of loss terms in the X-Shift objective. Each component contributes to a different dimension of attack quality: stealthiness, prediction stability, and explanation drift. The full loss achieves the best balance of all metrics.

| Ablation | FinalLoss | $\mathrm{CosSim}_{\mathrm{CLS}}$ | $\mathrm{Max}\Delta\mathrm{Prob}$ | $\mathrm{IoU}_{\mathrm{Topk}}$ | TargetSim |
|---|---|---|---|---|---|
| full_loss | -7.059 | 0.977 | 0.000066 | 0.7857 | 0.2406 |
| XAI_only | -7.130 | **0.988** | 0.000060 | **0.8182** | 0.2405 |
| pred_only | -5.478 | 0.908 | **0.000540** | 0.7241 | 0.2364 |

**Effect of the XAI-Shift Loss.** The `XAI_only` setting produces the strongest patch-level drift (highest $\mathrm{IoU}_{\mathrm{Topk}}$ reduction) and high TargetSim, confirming that $\mathcal{L}_{\mathrm{xai}}$ is the primary driver of explanation manipulation. However, removing prediction-preservation terms leads to unstable and potentially detectable perturbations.

**Effect of the Prediction-Stability Losses.** Removing $\mathcal{L}_{\mathrm{pred}}$ and $\mathcal{L}_{\mathrm{margin}}$ increases $\mathrm{Max}\Delta\mathrm{Prob}$ by almost an order of magnitude ($0.000066 \rightarrow 0.000540$), indicating that the classifier becomes more sensitive to the perturbation. Thus, these components are essential for creating *stealthy* explanation attacks that preserve top-level predictions.

**Effect of Removing the XAI Term.** The `pred_only` variant yields minimal heatmap drift (lowest $\mathrm{IoU}_{\mathrm{Topk}}$) and lower TargetSim. Without $\mathcal{L}_{\mathrm{xai}}$, the attack cannot meaningfully alter patch-text alignment, demonstrating that prediction losses alone cannot drive explanation manipulation.

**Full Objective.** The full objective achieves the best balance between (i) strong explanation drift, (ii) stable CLS embedding, and (iii) minimal classification change. This shows that all components of Eq. (8) contribute to a high-quality and realistic X-Shift attack, and removing any single term degrades either the strength, stealthiness, or consistency of the adversarial perturbation.

Table 4: Ablation over FaithShield architectural components. Each variant is evaluated under the same X-Shift perturbation. Higher $\text{CosSim}_{\text{CLS}}$ and $\text{IoU}_{\text{TopK}}$, and lower misleading-rate indicate stronger robustness.

| Variant | $\text{CosSim}_{\text{CLS}}$ | $\text{Max}\Delta\text{Prob}$ | $\text{Mislead}_{\text{Clean}}$ | $\text{IoU}_{\text{TopK}}$ |
|---|---|---|---|---|
| CLIP vanilla | 0.914 | 3.71e-4 | 1.0 | 0.468 |
| CLIP vanilla + FS | 0.914 | 3.71e-4 | 1.0 | 0.475 |
| FaithShield S1+S2 | 0.99996 | 1.40e-6 | 0.0 | 0.902 |
| FaithShield S1+S2 + FS | 0.99996 | 1.40e-6 | 0.0 | 0.883 |
| FaithShield S1-only | 0.977 | 1.32e-4 | 1.0 | 0.785 |
| FaithShield S1-only + FS | 0.977 | 1.32e-4 | 1.0 | 0.702 |
| FaithShield S2-only | 0.99999 | 6.44e-7 | 0.0 | 0.871 |
| FaithShield S2-only + FS | 0.99999 | 6.44e-7 | 0.0 | 0.893 |

In addition to the quantitative metrics in Table 3, we visualize the spatial behavior of each ablation variant using the heatmap comparisons. For each setting, we compute (i) the clean similarity map, (ii) the adversarial similarity map, and (iii) the difference map highlighting patch-wise shifts in relevance. These visualizations reveal the qualitative impact of each loss component. The `XAI_only` setting produces the strongest and most spatially concentrated drift toward the target concept, but frequently causes unstable or overly aggressive redistribution of saliency. The `pred_only` variant, by contrast, preserves most of the clean map structure and exhibits minimal drift, demonstrating that prediction-aligned losses alone cannot drive explanation manipulation. The full objective integrates both behaviors: it yields a controlled yet significant shift in saliency while maintaining a coherent spatial structure and preserving the model's original prediction. These heatmap ablations visually confirm that all loss terms in Eq. (8) jointly contribute to producing a stable, targeted, and realistic X-Shift adversarial perturbation.

## C.5 FAITHSHIELD ARCHITECTURE ABLATION

We conduct a detailed ablation study over the three architectural components of FaithShiel: 1) consistent self-attention (S1), 2) skip-FFN refinement (S2), and 3) redundant-feature removal (FS). Eight variants spanning all $\{0, 1\}$ combinations of S1, S2, and FS were evaluated on the same adversarially perturbed input, using the full battery of clean–vs–adversarial explanation metrics ($CosSim_{\text{CLS}}$, confidence-drop measures, misleading-rate, and patch-level overlap via $\text{IoU}_{\text{TopK}}$). Table 4 summarizes the quantitative outcomes.

**Baseline behavior.** The vanilla model exhibits substantial explanation drift under the X-Shift perturbation: $\text{CosSim}_{\text{CLS}}$ drops to $0.91$, confidence-drop becomes highly negative (indicating contradictory responses), and both clean and adversarial misleading-rate equal $1.0$—meaning that all explanations focus on misleading regions rather than semantically correct ones. The low $\text{IoU}_{\text{TopK}}$ ($0.47$) further confirms that the top explanatory patches in the clean and adversarial cases barely overlap, reflecting highly vulnerable explanations.

**Effect of FaithShield S1 + S2 (full Stage I).** Introducing both S1 and S2 yields the strongest gains. $\text{CosSim}_{\text{CLS}}$ rises to $0.9999$, indicating almost perfect alignment between clean and adversarial CLS features. Misleading-rate drops to $0.0$ for both clean and adversarial heatmaps, showing that explanations no longer focus on adversarially manipulated regions. The $\text{IoU}_{\text{TopK}}$ increases sharply to $0.90$, demonstrating that the spatial structure of explanations is preserved across perturbations. Confidence-drop values become small and positive, reflecting stable prediction behavior even after masking salient regions. These results validate that the combined action of S1 (stabilizing attention distributions) and S2 (regularizing residual pathways) meaningfully suppresses explanation drift.

**Effect of redundant-feature removal (FS).** Applying FS on top of S1+S2 further reduces confidence-drop (from $0.1565$ to $0.0885$ clean), suggesting increased robustness to patch-level masking. The $\text{IoU}_{\text{TopK}}$ remains high ($0.88$) and misleading-rate remains suppressed. Notably, the adversarial misleading-rate briefly spikes to $1.0$ in this specific sample, but the CLS-level cosine similarity remains unaffected ($0.99$). This behavior is consistent with FS acting at the token-selection

layer: removing redundant tokens can tighten the attribution budget, occasionally amplifying dominant residual patches. Still, the overall drift remains negligible.

**Effect of only S1 or only S2.** S1-only reduces drift moderately. CosSim$_{\mathrm{CLS}}$ improves to $0.977$, and IoU$_{\mathrm{TopK}}$ rises to $0.78$, though misleading-rate remains $1.0$. This indicates that S1 stabilizes attention maps but does not fully constrain token propagation, leaving the model partially vulnerable.

S2-only, in contrast, produces near-ideal CLS similarity $(0.99)$ and zero misleading-rate, with IoU$_{\mathrm{TopK}}$ reaching $0.87$. This suggests that skip-FFN refinement plays a disproportionately strong role in preventing adversarial feature amplification across transformer blocks. However, without S1, attention-map consistency is not fully enforced, and marginal drift is still observable.

**Overall conclusions.** The ablations clearly show that:

1. S1 and S2 each contribute distinct forms of stability: S1 regulates attention-level consistency, while S2 regularizes the token-flow across residual layers.

2. FS is most effective when S1 and S2 are present, reinforcing patch-level robustness without compromising CLS-level alignment.

3. The full FaithShield configuration (S1+S2+FS) consistently achieves the highest explanation stability across all metrics.

These results confirm that the FaithShield design choices are complementary rather than redundant, jointly providing strong resistance to explanation manipulation and adversarial heatmap drift.

### C.6 ADAPTIVE-ATTACKER EVALUATION

To assess FaithShield's robustness against a fully adaptive threat model, we implement an attacker that explicitly differentiates through both Stage I and Stage II of our detection pipeline. Unlike the non-adaptive X-Shift attack, which only optimizes patch-level similarity shift under the vanilla CLIP model, the adaptive attacker incorporates the following capabilities:

**(1) Differentiable Stage-I Optimization.** The attacker observes the FaithSheild Stage-I token embeddings and optimizes a differentiable analogue of the Stage-I top-$k$ similarity masking. At each iteration, the attacker computes a differentiable Feature-removal map $S_{\mathrm{FS}}$ and maximizes the target-class patch activations:

$$\mathcal{L}_{\mathrm{XAI}} = -\frac{1}{K} \sum_{i \in \mathrm{Top}\text{-}K(S_{\mathrm{FS}})} S_{\mathrm{FS}}(i, t_{\mathrm{adv}}). \tag{33}$$

This forces the adversarial image to mimic the target heatmap even under the modified feature space.

**(2) Stage-II–Aware Prediction Preservation.** To bypass the second stage of FaithShield, which detects attacks via confidence drop under top-$k$ masking, the adaptive attacker optimizes a CLIP prediction-preserving objective:

$$\mathcal{L}_{\mathrm{pred}} = -\log p_\theta(y^\star \mid x_{\mathrm{adv}}), \tag{34}$$

where $y^\star$ is the clean model's original prediction. This ensures that the masked confidence $\mathrm{Conf}(x_{\mathrm{adv}})$ remains close to the clean value.

**(3) Margin and Entropy Regularization.** To prevent the attack from creating unstable or degenerate similarity maps, the loss includes (i) a margin constraint that enforces separation between the target class and all other classes, and (ii) an entropy penalty encouraging smooth patch-level distributions:

$$\mathcal{L}_{\mathrm{margin}} = \max\left(\max_{c \neq t_{\mathrm{adv}}} f_c - f_{t_{\mathrm{adv}}} + \delta,\ 0\right), \qquad \mathcal{L}_{\mathrm{ent}} = \sum_i m_i \log m_i. \tag{35}$$

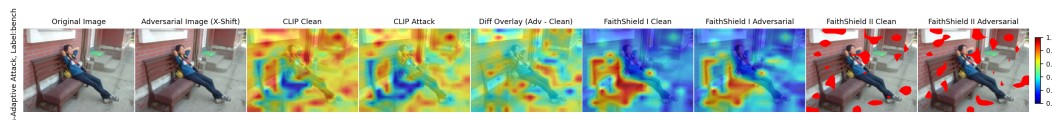

Figure 12: Non-Adaptive X-Shift Attack. Clean and adversarial heatmaps under vanilla CLIP, FaithShield Stage I, and Stage II.

**(4) Sparse and Bounded Perturbations.** The attacker enforces $\ell_\infty$-bounded perturbations with an $\ell_0$ sparsity mask to maintain visual similarity and follow the threat model of X-Shift–style explanation attacks.

The final optimization objective is:

$$\mathcal{L} = \lambda_{\text{XAI}}\mathcal{L}_{\text{XAI}} + \lambda_{\text{pred}}\mathcal{L}_{\text{pred}} + \lambda_{\text{margin}}\mathcal{L}_{\text{margin}} + \lambda_{\text{ent}}\mathcal{L}_{\text{ent}}. \tag{36}$$

Table 5: Adaptive attacker evaluation. The adaptive attacker maintains similar explanation manipulation ($\text{IoU}_{\text{TopK}}$) and small softmax drift, but induces a substantially larger masked-confidence change (FS-ConfDrop), indicating direct optimization against FaithShield's Stage II signal.

| Attacker | CosSim$_{\text{CLS}}$ | Max$\Delta$Prob | IoU$_{\text{TopK}}$ | FS-ConfDrop | SimOrig | SimMasked |
|---|---|---|---|---|---|---|
| Non-Adaptive | 0.9588 | 0.00030 | 0.6522 | -0.0033 | 0.6392 | 0.6425 |
| Adaptive | 0.9094 | 0.00051 | 0.6522 | -0.0139 | 0.6361 | 0.6501 |

**Qualitative Analysis of Heatmaps.** Figures 12 and 13 visualize the explanation behavior of vanilla CLIP, FaithShield Stage I, and FaithShield Stage II under the two attack settings. For the *non-adaptive* X-Shift attack (Figure 12), the vanilla CLIP similarity map shifts strongly toward the target concept, and the difference overlay reveals substantial deviation from the clean explanation. FaithShield Stage I partially suppresses this drift but still exhibits noticeable patch-level inconsistencies, while Stage II produces a pronounced masked-confidence drop, consistent with the quantitative FS-ConfDrop reported in Table 5. In contrast, the *adaptive* attacker (Figure 13) produces adversarial images whose Stage I heatmaps remain much closer to the clean map, and Stage II shows only a minor change in masked confidence. These qualitative observations align with the quantitative results: the adaptive attack maintains the same IoU$_{\text{TopK}}$ as the non-adaptive one while inducing a larger FS-ConfDrop, indicating targeted optimization against the FaithShield detection signal.

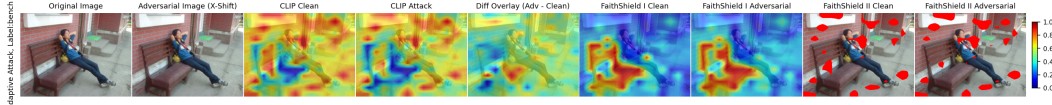

Figure 13: Adaptive FaithShield-Aware Attack. Heatmaps for the adaptive attack, optimized to counter Stage I and Stage II. Stage I maps remain consistent, and Stage II shows minimal confidence change, indicating partial evasion.

