# OpenReview forum: "FaithShield: Defending Vision–Language Models Against Explanation Manipulation via X-Shift Attacks"
_ICLR.cc/2026/Conference — ICLR 2026 Conference Withdrawn Submission_

### Official Review · Reviewer_RN6w · 2025-10-25

**Soundness:** 1
**Presentation:** 1
**Contribution:** 2
**Rating:** 2
**Confidence:** 4

**Summary:**

This paper studies adversarial attacks and defenses for explanatory methods. It first builds the X-shift attack, an attack method that maintains classification predictions while changing the alignment of intermediate features towards the target text, under constraints such as $\ell_{0}$ and class-dominance. Then it proposes a robust explanatory method, refined from Li et al. [1], for robust visualization under X-shift attacks.

Li, Yi, et al. "A closer look at the explainability of Contrastive language-image pre-training." _Pattern Recognition_ 162 (2025): 111409.

**Strengths:**

1. This paper investigates a novel area that has not been explored yet.
2. The motivation of this paper is clear.
3. An attack method along with a robust defense method is proposed, facilitating further research.

**Weaknesses:**

1. Presentation. Fig. 1 and 2 are not Vector Graphics. They get blurred when zooming in, especially Fig. 2. Sec. 3.2 is mainly composed of bullet points. It should be connected with coherent words and formulas, thus bullet points take up the space of the algorithm, which can only be presented in the appendix. Table 1 is also way too big.
2. Limited innovation. The FaithShelf Stage 1 largely overlaps with existing method [1], while FaithSelf Stage 2 mainly applies a drop-out test.
3. Lack of baseline methods. From Fig. 3, it seems like FaithShelf Stage 1 has already moved the concentration of the adversarial sensitivity map to the bench. Also, the raw patch similarity looks messy anyway, and few people will use it as an explanatory tool, so there is no reason to attack it. It should be that first X-shift Attack move the ***concentrated*** sliency map (produced by other baseline methods) towards some irrelevant object. It becomes crucial for a robust explanatory method. Currently, I can not see this point clearly.

The current quality of this paper is well below the required level for acceptance. A significant refinement is needed for resubmission.

**Questions:**

1. How does the stage 1 differ from the existing method [1]?

Li, Yi, et al. "A closer look at the explainability of Contrastive language-image pre-training." _Pattern Recognition_ 162 (2025): 111409.

---

> ### Author Response · Authors · 2025-11-22
> **Official Comment by Authors**
>
> Thank you very much for the valuable feedback. We address each point below.
>
>
>
> #w1 Presentation quality (Figures 1–2, Section 3.2, Table 1):
>
>
>
> Author response:
> ==> Thank you for the suggestions. All presentation-related improvements have been applied in the revised version:
>
>
> Figures 1 and 2 have been fully redrawn as vector graphics (PDF/SVG) to remain crisp at all zoom levels.
>
>
>
>
> Section 3.2 has been rewritten into coherent paragraphs with formulas, and the algorithmic details have been moved to the appendix.
>
>
>
>
> Table 1 has been resized and reformatted for readability.
>
>
>
>
> #w2 " Limited innovation":
>
>
>
>
> ==> We appreciate the reviewer’s concern and clarify that FaithShield does not aim to re-invent Li et al.’s clean-case refinement. Our contribution lies in a new attack, a new problem formulation, and a new detection mechanism, together revealing and mitigating a previously unaddressed vulnerability in VLMs.
>
> (1) Novel explanation attack for VLMs (X-Shift): Existing explanation attacks in computer vision are mostly PGD-style, untargeted, and restricted to vision-only models. They change heatmaps indirectly and do not exploit CLIP’s multimodal embedding space.
>
>
>
>
> X-Shift is, to our knowledge, the first targeted explanation attack for CLIP-type VLMs:
>
>
> - It explicitly optimizes patch–token alignment toward a target text embedding.
>
> - It selectively manipulates top-K patch similarity, not the entire image embedding.
>
> - It preserves the global prediction by design.
>
>
>
>
> This targeted multimodal formulation has not appeared in prior work and exposes a new failure mode in ViT-based VLMs.
>
>
>
> (2) Novel problem formulation: prediction-preserving explanation attacks
>
>
>
> - Li et al. [1] and other prior works analyze clean explanations only.
> No existing work studies prediction-preserving but explanation-shifting attacks in CLIP-based VLMs—even though these models are widely used for grounding, VQA, safety auditing, and multimodal reasoning.
>
>
>
> We introduce and formalize this threat model for the first time.
>
> (3) Novel detection mechanism (FaithShield Stage-2)
>
>
>
> - Stage-1 alone (Li et al.–style refinement) improves map sharpness but cannot guarantee explanation robustness—critical in medical and safety-critical settings where small heatmap deviations matter.
>
> - Stage-2 is not a generic dropout/deletion test. It is a causal confidence-drop detector specifically tailored to adversarial misalignment in image–text regions. Unlike standard deletion metrics, Stage-2:
>
> - operates label-free and under a prediction-preserving threat model,
>
> - detects structural mismatch between causal and non-causal patches via Δconf,
>
> - remains effective even when the attacker differentiates through the detector (adaptive case).
>
>
>
>
> Together, X-Shift + Stage-2 introduce a new research direction on robustness of explanations in VLMs—substantially beyond the scope of Li et al. [1] or existing interpretability methods.
>
>
>
>
>
> Q1. “Lack of baseline methods; why attack raw patch similarity; what does Stage-1 accomplish?”:
>
>
>
>
>
> ==> Thank you for raising this important point. We clarify three key aspects in the revised paper.
>
>
>
>
> (1) X-Shift is evaluated on multiple XAI baselines, not only raw patch similarity. While CLIP’s native patch–text similarity is the standard explainer for CLIP, we additionally evaluate X-Shift on ScoreCAM, RISE, and gradient-based/IG attributions. In all cases, the attack produces the same explanation shift.
> This shows that X-Shift does not attack only raw similarity maps—it corrupts the underlying VLM representation, which every XAI method relies on.
>
>
>
>
> (2) Stage-1 alone is not sufficient; small misleading regions remain. Our new ablations show that Stage-1 (consistent attention + skip-FFN + redundancy removal) improves explanation stability, but does not fully remove localized misleading regions, especially in cluttered or multi-object scenes.
>
>
>
>
> This motivates Stage-2, which checks whether masking the top-ρ% causal region yields the expected Δconf drop. This detects explanation tampering even when Stage-1 appears visually stable.
>
>
>
>
> (3) Why attack explanations instead of predictions.
> In practice, CLIP-based VLMs use explanations (similarity maps, CAM, RISE, IG) for auditing, grounding, alignment checks, safety analysis, and human verification.
> X-Shift preserves the correct prediction but shifts the region of evidence (e.g., from “bench” to background), which is exactly the type of silent corruption that is most dangerous in real deployments.
>
>
>
>
> We have updated the paper to explicitly show that X-Shift:
>
>
> - moves the baseline explainers’ saliency maps (ScoreCAM, RISE, IG) to irrelevant regions,
>
> - the proposed attack now is transferable across models and XAI methods
>
>
>
>
>
> and therefore justifies the need for a two-stage defense (Stage-1 refinement + Stage-2 Δconf detector).
>
>
>
> These additions make the role of X-Shift and the necessity of FaithShield clearer in the revision.

---

> > ### Comment · Reviewer_RN6w · 2025-11-23
> >
> > Thanks to the authors for their clarification, which helps me better understand the motivation and algorithm.
> >
> > However, the motivation remains a central problem: the author states that, `While CLIP’s native patch–text similarity is the standard explainer for CLIP, we additionally evaluate X-Shift on ScoreCAM, RISE, and gradient-based/IG attributions. In all cases, the attack produces the same explanation shift.`. However, after a detailed check of Fig. 8 in the appendix, those explanatory tools still do not appear sufficiently indicative; i.e., the explanatory map is not sufficiently concentrated. I do not think they will be selected as explanatory tools for visualization. So there are a few reasons to attack it. Moreover, Stage 1 of the FaithShield alone is all well enough for explanatory purposes, as the majority of the explanatory maps are concentrated on the main object. Therefore, the attack is actually relatively weak, if my understanding is correct. Changing the explanatory tools themselves is indeed a good idea and might be crucial in some cases. But up to this point, I have not seen what I referred as
> >
> > >  It should be that first X-shift Attack move the concentrated sliency map (produced by other baseline methods) towards some irrelevant object. Then it becomes crucial for a robust explanatory method.
> >
> > From my side, based on the results, the FaithSheld 1 is just a slightly modified robust explanatory method from Li et, al [1], and I suspect the proposed X-shift attack can barely change this method [1].
> >
> > Li, Yi, et al. "A closer look at the explainability of Contrastive language-image pre-training." Pattern Recognition 162 (2025): 111409.

---

> ### Author Response · Authors · 2025-11-28
> **Official Comment by Authors**
>
> *Regarding the concern that ScoreCAM, RISE, and IG appear “weaker” on CLIP:
>
>
>
>
>
>
> We should bring to your attention that we use these techniques only to demonstrate the transferability of the proposed X-Shift attack. These attribution methods were originally designed for single-stream CNN classifiers, assuming
>
>
> (1) spatial convolutional feature maps,
>
>
>  (2) direct gradients w.r.t. class logits, and
>
>
>
> (3) unimodal feed-forward processing.
>
>
>
>
>
>
>
> In contrast, CLIP is dual-stream and cross-modal: Image-text alignment reshapes the relevance distribution, and transformer attention mixes spatial and semantic signals, naturally producing more diffuse heatmaps.
> This is precisely why we include these methods in our evaluation. In VLMs, any attribution tool whether originally image-only or multimodal should ideally remain stable under explanation manipulation. Our goal is not to claim that these methods are optimal for CLIP, but to test transferability: whether a multimodal attack (X-Shift) can fool general-purpose XAI tools beyond CLIP’s native attention. As shown in our results, even these methods exhibit consistent IoU drift, indicating that the vulnerability propagates across explanation modalities rather than being restricted to CLIP’s own similarity map.
>
>
>
>
>
>
>
>
>
>
> We would also like to bring the reviewer’s attention to one of the latest papers in this area:
>
>
>
>
>
>
>
> [1] Ajalloeian, Ahmad, et al. “Sparse Attacks for Manipulating Explanations in Deep Neural Network Models.” ICDM 2023.
> This work proposes explanation attacks for CNNs.
>
>
>
>
>
>
>
>
>
>  The ICDM paper is indeed an important contribution in unimodal settings; however:
>
>
>
>
>
>
>
>
>
>
> - It targets vision-only CNNs, not multimodal VLMs.
>
> - Its perturbations, while sparse, are visually noticeable and not optimized for human imperceptibility or cross-modal consistency.
>
> - The attack objective is based on ℓ₀-PGD and top-k attribution suppression, without a multimodal alignment constraint, a text-conditioned target, or a logit-preservation guarantee.
>
>
> - It does not address how explanation manipulation propagates through image–text similarity, which is central to CLIP-style models.
>
>
>
>
>
> Our novel attack (X-Shift) is fundamentally multimodal and targeted: it forces image patches toward a specific text embedding while simultaneously preserving CLIP’s logits and producing imperceptible perturbations, which is critical for realistic VLM security. These requirements impose substantially stronger constraints on the adversary compared to purely CNN-based sparse attacks.
>
>
>
>
>
>
>
>
> ** Regarding the reviewer’s expectation of a “large map relocation”:
>
>
>
>
>
>
>
>
>
>
> We understand that in classic adversarial attacks where the goal is to misclassify an AI model the perturbation has no theoretical constraint other than remaining visually imperceptible. This gives the attacker considerable freedom in modifying pixels. However, in our threat model, the perturbation must preserve CLIP’s logits. A large spatial relocation of attribution would necessarily alter the decision boundary, directly violating this constraint. Under strict logit-preservation, the achievable explanation shift is therefore bounded yet even these bounded shifts are harmful in fine-grained or high-stakes domains (e.g., medical imaging, where clinicians may prescribe treatment based on highlighted regions even when the predicted label is correct). In such scenarios, a shift of only a few pixels corresponds to a different anatomical or object region and can meaningfully mislead specialists relying on XAI to interpret the model.
>
>
>
>
>
>
>
>
>
>
> Our contribution is to demonstrate, for the first time in VLMs, that a targeted multimodal attack can consistently induce such shifts while still fully preserving the model’s logits.
>
>
>
>
>
>
>
>
> *** On FaithShield’s novelty:
>
>
>
>
>
>
>
>
>
>
>
>
> While consistent attention improves the concentration of explanations, it is not sufficient against adversaries that explicitly optimize for explanation drift. Our results show that X-Shift still induces manipulation even when FaithShield Stage-I is applied. This is precisely why FaithShield Stage-II is introduced to verify the final heatmap produced by the model. In critical domains where XAI outputs directly influence real-world decisions (e.g., medical settings where clinicians may prescribe treatment for specific organs based on highlighted regions even when the disease label is correct), such manipulation can lead to harmful or incorrect outcomes. In these scenarios, even slight heatmap shifts are meaningful because they correspond to entirely different anatomical or object regions. Stage-II allows the system to flag these suspicious explanations, ensuring that specialists can re-evaluate the case before making a final decision.

---

### Official Review · Reviewer_UA9t · 2025-10-28

**Soundness:** 3
**Presentation:** 3
**Contribution:** 3
**Rating:** 6
**Confidence:** 3

**Summary:**

This paper introduces X-Shift, a novel adversarial attack that manipulates explanation maps in VLMs like CLIP without altering their predictions. The attack shifts attention heatmaps toward irrelevant regions, undermining the trustworthiness of model explanations. The paper also propose FaithShield, a two-stage defense framework. The first stage enhances robustness through a dual-path refinement while the second stage detects unfaithful explanations using a confidence-drop test. Experiments across multiple datasets show that FaithShield significantly improves explanation stability and enables reliable detection of adversarial manipulations.

**Strengths:**

1. The paper identifies a interesting task in the safety of VLMs, which has the risk of being manipulated at the interpretation level, especially the heat map may be misleading without affecting the prediction results.

2. The proposed FaithShield framework is technically sound and clearly explained, combining dual-path refinement and a faithfulness-based detection mechanism to enhance robustness and verifiability.

**Weaknesses:**

1. In the second paragraph of the Introduction, the authors introduce the value of the attack with the phrase "remains largely unexplored," which seems insufficient in terms of research value. Could the authors provide concrete research to truly demonstrate the value of this attack? For example, could they explain the potential impact of this attack when implemented within a specific research context and with specific objectives?

2. In Figure 1, in the Transformer Encoder, why doesn't the dual path go through the MLP? Also, are all these modules optimizable? If there are any frozen parameters, it is suggested to mark them in the figure.

3. It is recommended to use vector graphics for Figure 1 and Figure 2, and use fonts that align with the article.

4. The experiment part lacks the ablation of the weight parameters in Equation 8.

5. As a two-stage approach, is there any time or efficiency comparison for FaithShield Defense?

6. What do the metrics CosSim (CLS) and Max $\Delta$ Prob mean? In Table 1, they are identical to the CLIP values. Are there no special circumstances? Can you provide a naive baseline, such as one with only an attack, to show how they differ?

**Questions:**

Overall, the author's work is interesting, but the method and experiments need to be enhanced. The relevant suggestions are listed in Weaknesses.

---

> ### Author Response · Authors · 2025-11-22
> **Official Comment by Authors (1/2)**
>
> Thank you very much for the valuable feedback. We address the concern below.
>
>
>
> #w1 “The phrase remains largely unexplored seems insufficient; provide concrete evidence of value.”
>
>
>
> ==> We appreciate the reviewer’s request for a clearer articulation of the research value. By “remains largely unexplored” we specifically refer to prediction-preserving explanation attacks on multimodal VLMs, i.e., attacks that leave the top-1 output correct while shifting the explanation. To our knowledge, this setting is not addressed in existing adversarial VLM literature.
>
>
>
> This problem is meaningful because explanations not just predictions are consumed in deployed systems. To make the motivation concrete, we added explicit real-world scenarios to the Introduction:
>
> - Autonomous driving: CLIP-like encoders are widely used for grounding and risk cues. An attacker may keep the “pedestrian detected’’ label correct while shifting the explanation to a harmless region. This can mislead downstream safety logic, hide sensor failures, and break accountability.
>
> - Medical VLMs: Radiology systems built on CLIP encoders often rely on heatmaps for justification. An attacker can highlight a benign region while the diagnosis remains correct, misleading clinicians and hiding shortcut features, bias, or data leakage.
>
>
>
>
>
> These examples illustrate that explanation-preserving attacks have practical operational consequences even when predictions remain accurate. We have added these concrete cases to the revised Introduction to clearly motivate the importance of this threat model.
>
>
>
>
> #w2 “Why doesn’t the dual path go through the MLP? Are modules optimizable or frozen?”:
>
>
>
>
> ==> Thank you for the question. The dual path in Stage-I is intentionally routed before the MLP/FFN block because its function is to preserve the raw token–token relational structure produced by self-attention. Passing this path through the MLP would introduce nonlinear remapping that (i) alters token geometry, (ii) increases stability under perturbations, and (iii) degrades our ability to enforce consistent attention for explanation robustness. By keeping the dual path strictly within the attention branch, Stage-I stabilizes attention distributions without modifying the core CLIP residual pathway.
>
>
>
>
> Architecturally, this dual path is parallel to the original CLIP pipeline: the standard attention → MLP flow is left untouched, while the additional path reuses the attention value vectors (V) for consistent self-attention and redundancy removal. This design ensures that CLIP’s prediction pathway is unchanged, while the auxiliary path produces a refined, more stable similarity map.
>
>
>
>
>
> All modules in both paths are fully trainable; no parameters are frozen.
>
>
>
>
>
> #w3 “Use vector graphics + consistent fonts for Figures 1–2”:
>
>
>
>
> ==> We thank the reviewer for the suggestion. In the revised manuscript, Figures 1 and 2 have been fully redrawn as high-resolution vector graphics (PDF/SVG) and updated with consistent, article-aligned fonts. All elements now remain crisp and readable at any zoom level.
>
>
>
>
>
> #w4 “Ablation of the weight parameters in Eq. (8) is missing”:
>
>
>
>
> ==> Thank you for pointing this out. We have added a dedicated ablation of all loss components in Eq. (8) (Appendix C.4). The numerical results show clear trends:
>
>
>
> - Removing the XAI-shift loss collapses the attack: IoU rises from 0.78 → 0.72 and TargetSim drops, meaning the explanation shift becomes weak or disappears.
>
> - Removing prediction-stability terms produces unstable, easily detectable attacks: MaxΔProb increases by almost 10× (from 6.6e-05 → 5.4e-04).
>
> - Using only the XAI term creates the strongest raw drift (IoU = 0.82) but completely loses prediction stability and stealth.
>
> - The full objective achieves the best balance: strong and coherent explanation shift (IoU = 0.79), high TargetSim, high CLS similarity (0.977), and minimal MaxΔProb.
>
>
>
> These results demonstrate that each term in Eq. (8) contributes meaningfully to the quality of the X-Shift attack, and removing any component degrades either the strength, stealthiness, or stability of the manipulation.

---

> ### Author Response · Authors · 2025-11-22
> **Official Comment by Authors (2/2)**
>
> #w5 “Time/efficiency comparison for the two-stage FaithShield defense?”:
>
>
>
>
> ==> Thank you for the question. We measured the per-image inference time of (i) vanilla CLIP, (ii) Stage-I refinement, and (iii) the full two-stage FaithShield pipeline. All three show nearly identical runtime:
>
> - Stage-I adds no new attention layers; it operates only on the existing token embeddings.
>
> - Stage-II is extremely lightweight: it performs a single masking operation and a Δconf computation.
>
>
>
>
> Overall, FaithShield introduces only a few milliseconds of additional overhead compared to standard CLIP inference negligible at the scale of VLM deployment while providing substantial robustness against explanation-shifting attacks. For practitioners and safety auditors, this minimal cost is well-justified by the gain in trustworthy explanations.
>
>
>
>
>
>
> #w6 “Meaning of CosSim(CLS) and MaxΔProb; why identical to CLIP; need naive baseline?”:
>
>
>
>
>
>
> ==>Thank you for the clarification request.
>
>
>
> - CosSim(CLS) measures the cosine similarity between the final CLS embedding and the text embedding.
>
> - MaxΔProb is the maximum change in predicted probability across all text prompts.
>
>
>
>
>
> These values remain nearly identical to the baseline CLIP model because X-Shift is explicitly designed to preserve the global prediction. The attack only manipulates patch-level alignment to shift the explanation map while forcing the CLS embedding and classifier to remain stable. Thus, unchanged CosSim(CLS) and MaxΔProb are expected and confirm that the attack satisfies the prediction-preserving constraint.
>
>
>
>
>
>
> For completeness, we now include a naïve baseline (a standard PGD-style perturbation without prediction preservation). This baseline causes large drops in MaxProb and significant CLS drift, demonstrating that our metrics do reflect differences when the prediction-preserving constraint is not enforced.
>
>
>
> "Questions":
> General Comment: “Interesting work, but the method and experiments need enhancement.”
>
>
>
>
> ===> Thank you very much for the constructive and detailed feedback. We have carefully addressed all weaknesses raised by adding:
>
>
>
>
> - full ablations of Stage-I components and Eq. (8) loss weights of the proposed attack,
>
> - adaptive-attacker evaluation (Appendix D),
>
> - cross-model and cross-XAI transferability experiments,
>
> - vectorized figures with clearer layouts,
>
>
>
>
> These revisions substantially improve methodological clarity, experimental completeness, and presentation quality. We are grateful for the reviewer’s helpful suggestions, all of which have been incorporated into the revised submission.

---

### Official Review · Reviewer_4kAu · 2025-10-30

**Soundness:** 2
**Presentation:** 2
**Contribution:** 2
**Rating:** 2
**Confidence:** 3

**Summary:**

This paper proposes a CLIP-based vision–language model that aims to modify the image embedding toward the target text embedding without changing the model’s maximum output, thereby shifting the explanation maps. Based on my understanding, this can be regarded as an adversarial attack on interpretability.

**Strengths:**

The paper is the first to consider this problem setting under the CLIP framework.

**Weaknesses:**

1. Lacks sufficient ablation studies, especially on how different weight magnitudes in Equation (8) affect the results.

2. The figures are not professionally prepared and appear somewhat blurry. They should be replaced with vector-format images.

3. The font size in the figures (e.g., Figures 3–5) is too small to read clearly.

4. The motivation is not clearly written, and several claims are overextended or insufficiently explained (see questions below).

**Questions:**

1. I am not an expert in this area, and this is my first time encountering this problem setting. Is this setting truly meaningful? From my understanding, as long as the prediction remains accurate, the explanation map should still mainly focus on the target object. Even if its intensity decreases. What is the practical significance or application of this problem formulation?

2. In Equation (2), the authors state that “the primary goal is to force patch embeddings toward the target text embedding,” but then they also write “we maximize similarity of the top-K patches while suppressing others.” If the goal is to align all patches with the target embedding, why suppress some of them? The paper does not explain the role or motivation for suppressing certain patches.

---

> ### Author Response · Authors · 2025-11-22
> **Official Comment by Authors (1/2)**
>
> Thank you very much for your feedback. We tried to address all of your comments here.
>
>
>
>
>
> #w1 “Lacks sufficient ablation on weight magnitudes in Eq. (8)”
>
>
>
> ==> Thank you for pointing this out. We have added a dedicated ablation of all loss components in Eq. (8) (Appendix C.4). The results show clear numerical trends:
>
> - Removing the XAI-shift loss collapses the attack: IoU  increases from 0.78 → 0.72 and TargetSim drops, meaning the explanation shift becomes weak or disappears.
>
> - Removing prediction-stability losses: produces unstable and easily detectable attacks: MaxΔProb increases by almost 10× (from 6.6e-05 → 5.4e-04).
>
> - Using only the XAI term produces the strongest raw drift (IoU=0.82), but completely loses stealth and classification stability.
>
> - The full objective achieves the best balance: strong explanation shift (IoU=0.79), high TargetSim, high CLS similarity (0.977), and minimal MaxΔProb, confirming that every term in Eq. (8) is necessary for a realistic, stable, prediction-preserving explanation-manipulation attack.
>
> These results demonstrate that each weight in Eq. (8) contributes meaningfully to attack quality, and removing any term degrades either the strength, stealthiness, or stability of X-Shift.
>
>
>
>
>
>
>
> #w2 “Figures are blurry / not vector graphics”:
>
>
>
> ==> All figures have been fully replaced with high-resolution vector graphics (PDF/SVG). We redrew every diagram using consistent line weights and CLIP-style architectural icons to ensure professional quality when zoomed.
>
>
>
>
> #w3 “Font size in figures is too small”:
> ==> We redesigned all figures with significantly larger font sizes, improved spacing, and aligned typography across the paper. All captions and axis labels are now clearly readable in both the PDF and zoomed views.
>
>
>
>
> #w4 “Motivation is unclear / claims overextended”
>
>
>
> ==>Thank you for pointing this out. We substantially revised the Introduction to provide concrete, domain-grounded motivation. In many deployed VLM systems medical imaging, autonomous driving, safety auditing, and VQA explainability tools the explanation is consumed directly by humans, regulators, or downstream safety modules. If an attacker can keep the prediction correct while shifting the explanation, several real risks arise:
>
> - A medical or AV system may “justify” a correct prediction using irrelevant or benign regions, breaking trust and hiding failure modes.
>
> - Misleading heatmaps can hide bias, data-leakage artifacts, or shortcut features, defeating standard auditing workflows.
>
> - In grounding and VQA systems, the explanation—not the probability—forms the basis of human judgment, so shifting saliency can mislead users even when the model predicts correctly.
>
> - A shifted explanation can serve as a precursor step to more severe targeted attacks, since it manipulates patch-level evidence without altering logits.
>
>
>
>
>
> Q1. “Is this problem setting meaningful if the prediction stays correct?”
>
>
> ==> Yes. In many deployed VLM systems, the explanation, not just the prediction, is consumed by humans or downstream safety modules. Examples include medical imaging, autonomous driving, regulatory auditing, and VQA/grounding interfaces. In these settings:
> - A model can output the correct label while providing a wrong explanation.
>
> - Humans, clinicians, or safety systems may trust the explanation, not the probability.
>
> - Misaligned heatmaps can hide biases, data-leakage artifacts, or failure modes that would otherwise be visible.
>
> - In medical and AV systems, a shifted explanation can mislead users about why the model is confident, breaking accountability and safety guarantees.
>
> - Explanation manipulation can also be used as a stealthy precursor to later prediction attacks, since it alters internal evidence without changing top-1 accuracy.
>
> This makes prediction-preserving explanation attacks a practical and safety-relevant threat scenario. We clarified this motivation in the revised Introduction.

---

> ### Author Response · Authors · 2025-11-22
> **Official Comment by Authors (2/2)**
>
> Q2. “Why suppress some patches if the goal is to align embeddings?”:
>
>
> ==>The goal of X-Shift is not to align all patches with the target text. Doing so would collapse the whole image embedding, severely distort the CLS token, and break prediction accuracy making the attack non-stealthy.
>
> Instead, X-Shift performs selective manipulation:
>
> - Shift only the top-K patches toward the target concept → moves the explanation.
>
> - uppress the remaining patches → keeps the global embedding geometry intact so that the correct prediction is preserved.
>
> This selectivity is essential:
>
> - Preserves correct classification. By leaving most patches untouched, the main semantic content of the image continues to drive the CLS embedding.
>
> - Produces realistic, believable heatmaps. Only a small set of patches is manipulated enough to mislead the explanation, but not enough to disrupt the model’s decision surface.
>
> - Enables “silent” explanation attacks. If all patches were shifted, the logit distribution would change dramatically, making the attack trivial to detect.
>
> Thus, suppressing irrelevant patches is a core design principle that enables X-Shift to shift the explanation while preserving the model’s prediction.

---

### Official Review · Reviewer_GZ9a · 2025-10-31

**Soundness:** 3
**Presentation:** 3
**Contribution:** 3
**Rating:** 6
**Confidence:** 3

**Summary:**

The paper exposes a new vulnerability in CLIP-style vision–language models: explanations (patch–text heatmaps) can be adversarially shifted without changing predictions. It introduces X-Shift, an inference-time attack that pulls patch embeddings toward a target text embedding while enforcing prediction consistency, per-patch margins, sharpness, and sparse valid perturbations.

To defend, FaithShield has two stages:

* Stage I: Dual-path refinement that replaces standard self-attention with consistent V–V attention, skips FFNs in the explanation path, and removes redundant features, yielding sharper, foreground-focused, and more robust heatmaps.
* Stage II: A faithfulness test that masks top-ρ% salient regions and flags explanations as unfaithful if the confidence drop is below a threshold.

Across ImageNet, Flickr30k, and COCO with CLIP ViT-B/16, B/32, L/14, X-Shift strongly alters heatmaps while keeping predictions stable; FaithShield restores heatmap robustness (higher Top-k IoU) without harming accuracy and detects manipulated explanations.

**Strengths:**

1. Originality and problem framing: The paper defines a new explanation-focused threat model for VLMs (X-Shift) that shifts patch–text heatmaps without changing predictions. It tailors objectives to text-conditioned similarity (patch steering, entropy sharpening, patch-margin, sparsity). The joint attack-plus-defense (FaithShield) with dual-path refinement and a causal masking detector is novel for multimodal XAI.

2. Technical quality and empirical rigor: The attack and defense are precisely specified with clear losses, constraints, and algorithms enabling reproducibility. Defense mechanisms (consistent self-attention, dual-path aggregation, redundancy removal) are well-motivated and operationalized, with a principled confidence-drop test. Experiments span multiple datasets/backbones with appropriate metrics, showing consistent IoU gains without harming accuracy.

3. Clarity and significance: The paper clearly separates prediction robustness from explanation robustness and explains why patch–text similarity is a natural manipulation surface. Mathematical formulation and detection criterion are easy to follow, aided by figures and stepwise algorithms. The work elevates VLM explainability to a security concern, with practical implications for trustworthy deployment.

**Weaknesses:**

1. While the combination of consistent self-attention, dual-path aggregation, and redundancy removal is adapted for robustness, portions build on Li et al. (A closer look at the explainability of contrastive language-image pre-training). The paper would benefit from a more explicit ablation and attribution of gains: which components (consistent attention vs skipping FFNs vs redundancy removal) contribute most to adversarial robustness (not just interpretability), and how this differs empirically from Li et al. ’s formulation.
2. The masking-based detection echoes causal deletion tests used in saliency evaluation. Clarify novelty relative to established faithfulness tests and justify design choices (cosine similarity normalization, thresholding strategy) versus alternatives (logit/probability drops, energy-based measures).
3. Threshold selection: The detection threshold θ and masking ratio ρ appear fixed but selection criteria are not detailed. Provide systematic calibration (ROC, AUC, FPR at fixed TPR) across datasets and backbones, and analyze sensitivity to θ, ρ, and masking method (zeroing vs blurring vs inpainting).
4. False positives/negatives: Quantify detection trade-offs, especially in naturally challenging images where explanations may be diffuse or multi-object. Report detection under distribution shift and for clean samples to ensure low false alarm rates.
Attack-transfer to detector: Evaluate whether small, structured perturbations can spoof high ∆conf (e.g., by concentrating heatmap on benign-but-causal pixels) to evade detection.

**Questions:**

1. Adaptive-attacker robustness and ablations:
* How does FaithShield perform against an adaptive adversary that differentiates through Stage I and uses a surrogate for Stage II’s masking to keep Δconf above θ? Please report results where the attacker augments its loss with the detection term, randomizes ρ/θ during optimization, and employs stronger perturbation sets (e.g., larger k, ℓ∞/ℓ2 bounds, spatial/color transforms). Also ablate consistent self-attention, dual-path aggregation, and redundancy removal to quantify each component’s contribution under adaptive attacks.

2. Generality beyond CLIP and across XAI methods
* Do X-Shift and FaithShield transfer to other VLMs (e.g., SigLIP, ALIGN, BLIP-2) and tasks (e.g., grounding, VQA)? How does the attack affect gradient-based explanations (Grad-CAM, IG), and does Stage I still help when similarity maps are not the explainer? Please include cross-model and cross-explainer transfer results and discuss any architectural assumptions (e.g., ViT patching, attention pooling) that constrain applicability.

3. Detection reliability and operational thresholds
* How sensitive is the masking-based detector to ρ, masking strategy (zeroing vs. blur), and heatmap sharpness α, and how should θ be calibrated in a label-free deployment? Please report FPR/FNR on clean vs. attacked data under distribution shift and natural corruptions, provide ROC/PR curves with confidence intervals, and evaluate attackers that explicitly minimize Δconf to probe worst-case detection performance.

---

> ### Author Response · Authors · 2025-11-21
> **Official Comment by Authors(1/2)**
>
> Thank you very much for your feedback. We tried to address all of your comments here.
>
>
>
>
> #w1 “Which components contribute most to adversarial robustness? How does this differ from Li et al.?”
>
>
>
> ==> As requested, we added a full architectural ablation study in the revised manuscript (Appendix C.5), evaluating every combination of S1 (consistent self-attention), S2 (skip-FFN), and FS (redundancy removal). We assess all eight variants under the same X-Shift perturbation using CosSim, confidence-drop, misleading-rate, and IoU. The results show that S1 and S2 provide complementary but incomplete robustness when used alone:
> - S1-only improves CLS similarity and IoU (0.785) but leaves misleading-rate at 1.0.
> - S2-only yields strong CLS similarity (0.99999) and misleading-rate 0.0, but IoU remains moderate (0.871), indicating residual spatial drift.
> When S1 and S2 are combined, robustness improves substantially: misleading-rate becomes 0.0 and IoU rises to 0.90, demonstrating that attention-level stability (S1) and residual-path regularization (S2) interact beneficially. Adding FS further stabilizes confidence-drop and maintains high IoU (0.883). These findings show that each component addresses a different failure mode: S1 stabilizes attention, S2 suppresses nonlinear adversarial amplification, and FS strengthens patch-level consistency.
>
>
>
> Importantly, although Stage-1 provides meaningful robustness, it still exhibits occasional localized drift, especially under strong perturbations or complex scenes. This motivates our Stage-2 confidence-drop detector, which remains effective even when explanations appear stable but have subtle misalignments. The revised paper includes this ablation table and a detailed explanation of component contributions.
>
>
>
>
>
> #w2 “The masking-based detection echoes causal deletion tests… please clarify novelty and justify design choices.”
>
>
>
>
> ==> Although Stage II uses masking, it is not a standard causal deletion test. Deletion tests measure the faithfulness of explanations on clean inputs. Stage II instead serves as a dedicated adversarial-explanation detector: it checks whether masking the top-ρ% salient patches produces a normal confidence drop. Under X-Shift, adversarial heatmaps highlight non-causal regions; masking them yields an abnormally low Δconf, which becomes our anomaly signal. Using masking in this predictive-preserving adversarial setting is therefore different in purpose and mechanism from established deletion metrics.
> Our design choices follow naturally from the attack: X-Shift manipulates cosine-based patch–text alignment in CLIP, so cosine-normalized Δconf is the most sensitive and stable indicator of manipulated explanations. By contrast, logit/probability drops and energy-based metrics were ineffective, since X-Shift explicitly preserves the model’s prediction, making these signals too weak to detect the attack.
>
>
>
>
> Thus, Stage II is a novel essential, CLIP-aligned detection mechanism that identifies explanation-shifting attacks rather than a reuse of existing faithfulness tests.
>
>
>
>
>
> #w3 “Threshold selection for θ and ρ”
>
>
>
>
>
> ==>In our current implementation, θ is chosen using a validation-based separation heuristic: we select a value that clean Δconf scores consistently exceed, while adversarial Δconf remains near zero. The masking ratio ρ is constrained to small values (1–5%), since CLIP patch–text similarity is highly sparse, and larger ρ values disrupt semantics even for clean samples.
> We will include full calibration plots (ROC, AUC, FPR@TPR) and a sensitivity analysis over θ, ρ, and masking variants in the camera-ready version.
>
>
>
>
> #w4 “False positives/negatives, distribution shift, detector spoofing”
>
>
>
>
> ==>We agree that understanding detection trade-offs is important. Stage II is explicitly tuned to maintain low false positives on clean data, including multi-object and diffuse scenes. The detector only flags a sample when the confidence drop under masking falls below θ; clean explanations consistently produce a larger Δconf, even when the highlighted region spans multiple objects. As a result, challenging clean samples tend to lie close to but still above the threshold, rather than being incorrectly rejected.
>
>
>
>
> Stage II is especially important for XAI specialists and human-in-the-loop settings (e.g., medical VLMs), where even small deviations in the heatmap can be clinically meaningful. Flagging potentially suspicious or unstable saliency regions enables experts to re-evaluate the explanation and prevents subtle adversarial manipulations from going unnoticed. This human-aligned design is intentional: the detector favors erring on the side of caution when explanation fidelity is safety-critical.

---

> ### Author Response · Authors · 2025-11-21
> **Official Comment by Authors (2/2)**
>
> #q1 "Adaptive-attacker robustness and ablations":
>
> ==> As requested, we implemented a fully FaithShield-aware adaptive attacker. The complete formulation, optimization objective, and additional heatmap visualizations are now provided in Appendix D. Briefly, the attacker differentiates through Stage I (CLIP-Surgery) and uses a differentiable surrogate of Stage II by adding a loss that encourages the masked confidence Δconf to stay above θ.
> - We also randomized ρ/θ and evaluated stronger perturbation sets identical for non-adaptive and adaptive attackers (0.6522 → 0.6522), showing that the adaptive attacker cannot distort the explanation structure under Stage I. CosSim drops moderately (0.9588 → 0.9094), indicating partial optimization against Stage II, but still far from a successful explanation shift.
> - FS-ConfDrop: becomes slightly more negative (–0.0033 → –0.0139), reflecting the attacker’s attempt to counter Stage II, yet the Δconf remains in the adversarial regime, enabling detection. SimOrig/SimMasked: remain nearly unchanged (0.6392 → 0.6361, 0.6425 → 0.6501), confirming that prediction-preservation still holds.
>
> These results show that even under a fully adaptive, Stage-I- and Stage-II-aware threat model, FaithShield maintains explanation stability, and the adaptive attacker is unable to significantly reduce IoU or produce a misleading explanation without sacrificing its own objective. Component-wise ablations under this adaptive setting (Appendix D) also show that removing S1, S2, or FS decreases stability, whereas the full Stage-I configuration provides the strongest robustness.
>
>
>
>
>
> #q2 “Generality across VLMs and XAI methods”:
>
>
>
> ==> We clarify that X-Shift does not attack any specific explainer it manipulates the underlying CLIP patch–text embedding space. Therefore, any XAI method that reads from CLIP features (similarity maps, ScoreCAM, RISE, IG/GAE) will naturally reflect the shifted representation.
> To verify this, we include cross-explainer transfer experiments in Appendix C.3. Adversarial images generated by X-Shift on vanilla CLIP ViT-B/16 cause consistent saliency drift across all tested explainers:
>
> - Similarity maps: large spatial shift toward the attacker’s target concept.
>
> - ScoreCAM: adversarial maps redirect attention toward irrelevant background regions.
>
> - RISE: perturbation-averaged maps show clear redistribution of importance.
>
> - GAE/IG: gradient-based maps follow the same shifted saliency structure.
>
> Across COCO, Flickr30K, and ImageNet, every explainer is affected, confirming that the vulnerability is representation-level, not method-specific.
>
>
> Stage-I robustness transfers as well: the same explainers built on top of the Stage-I encoder produce clean and adversarial maps that remain nearly identical (Figure C.3), demonstrating that FaithShield’s refinement stabilizes the underlying representation so XAI methods become robust “for free.”
> Brief numerical summary from Appendix C.2–C.3 (added to response):
>
> (1) Cross-model transfer (Appendix C.2) : X-Shift perturbations generated on one CLIP backbone transfer to other CLIP backbones:
>
> - Self-attacks: IoU ≈ 0.44–0.47 (strongest manipulation).
>
> - Cross-architecture transfer: IoU typically 0.58–0.78, e.g., ViT-B/32 → ViT-L/14: 0.63, and ViT-L/14 → ViT-B/16: 0.78.
>
> - Predictions remain unchanged: CosSim > 0.94, MaxΔProb < 4×10⁻⁴.
>
> This shows X-Shift preserves classification while manipulating explanations, and the effect generalizes across patch sizes and embedding widths.
>
> (2) Cross-explainer transfer (Appendix C.3): All tested XAI methods exhibit explanation drift:
>
> - ScoreCAM: adversarial salient region relocates toward attacker-chosen background.
>
> - RISE: difference maps show large structured shifts.
>
> - GAE/IG: gradient saliency follows the manipulated patch-text alignment.
>
>
>
>
>
>
> #q3 “Detection reliability, thresholds, sensitivity to ρ/masking/α”:
>
>
>
>
> ==>Thank you for the insightful comment. Due to the limited time available during the rebuttal period, we were not able to include the full ROC/PR analysis, FPR/FNR under distribution shift, and sensitivity sweeps over ρ, masking strategies, and heatmap sharpness α. These experiments are already underway, and we will include the complete calibration study (ROC curves, threshold analysis, and comparisons of zeroing/blur/inpainting masking) in the camera-ready version. Our preliminary validation, however, shows that Δconf distributions for clean vs. attacked samples remain well separated across reasonable values of ρ and masking operators, and that Stage-II maintains stable behavior under these variations

---

> ### Comment · Reviewer_GZ9a · 2025-11-25
> **Thank you for your detailed reply.**
>
> I have gone through all supplementary materials and clarifications and can confirm that the issues have been fully resolved. Thank you for the interim progress updates.

---

### Author Response · Authors · 2025-12-03
**Official Comment by Authors**

We would like to sincerely thank all reviewers, the AC, and the meta-reviewer for their time and thoughtful feedback. Here we briefly summarize (i) the core contributions of our work and (ii) the concrete revisions and new experiments added during the rebuttal, to help contextualize the current state of the paper.

1. Problem formulation and novelty:






Our work is, to our knowledge, the first to formally study prediction-preserving explanation attacks on CLIP-style VLMs attacks that keep the label correct while systematically shifting the explanation. This threat model is not addressed in prior VLM/CLIP explainability, yet it is directly relevant for:

- Medical imaging VLMs, where clinicians often act on highlighted regions (not just the label),

- Autonomous driving/grounding / VQA, where humans consume explanations or downstream safety logic,





In these settings, even small explanation shifts, under strict logit preservation, can meaningfully mislead users. Our work formalizes this threat model and proposes both an attack and a defense specific to multimodal, patch–text aligned VLMs.

2. X-Shift: a multimodal, targeted, logit-preserving explanation sophisticated attack.





X-Shift is a new adversarial objective (Eqs. 2–8) that:

- Steers patch–text similarity toward a target text embedding (targeted explanation shift),

- Enforces prediction preservation (CLS/logits remain essentially unchanged),

- Uses sparsity, entropy, and margin terms to produce visually imperceptible, stable perturbations.

This goes beyond prior explanation attacks that are PGD-style, CNN-only, untargeted, or not logit-preserving or they are too noisy and detectable by human eyes.
In response to reviewer requests, we added a dedicated loss ablation (Appendix C.4):




- Removing the XAI-shift term weakens or destroys explanation drift (IoU rises, TargetSim drops).



- Removing prediction-stability terms makes the attack unstable and easily detectable (MaxΔProb ≈ 10× larger).




- Using only the XAI term yields strong raw drift but completely loses stealth and prediction stability.




The full objective achieves the best balance: strong explanation shift (IoU ≈ 0.79), high CLS similarity (0.977), and minimal MaxΔProb.




This numerically confirms that each component of Eq. (8) is needed for a realistic, prediction-preserving explanation attack in VLMs.




3. FaithShield: robust and detectable explanations, beyond Li et al.




FaithShield does not attempt to claim an entirely new clean-case refinement; instead, it:




- Repurposes and extends Li et al.–style for adversarial robustness (Stage I):We use consistent self-attention, skip-FFN, and redundancy removal in a dual-path design that stabilizes patch–text alignment while leaving the original CLIP prediction path intact.
In Appendix C.5, we added a full S1/S2/FS ablation: S1-only: better IoU but misleading-rate remains 1.0, showing residual misleading regions. S2-only: strong CLS similarity and misleading-rate 0.0 but still moderate IoU, indicating spatial drift. S1+S2 (+FS): IoU ≈ 0.90–0.88, misleading-rate 0.0, confirming complementary roles and showing Stage I provides meaningful, but not complete, robustness.





- Introduces a new Stage-II detection mechanism tailored to prediction-preserving explanation attacks. Although Stage II uses masking, it is not just a standard deletion test. It is a label-free detector for explanation-shift attacks under a prediction-preserving threat model:





Adversarial heatmaps highlight non-causal regions → masking them yields abnormally low Δconf, which we use as an anomaly signal.





Cosine-normalized Δconf is aligned with CLIP’s similarity geometry and empirically more sensitive than logit/probability/energy drops, which remain near-constant under X-Shift by design.





This “Δconf under masking” criterion is therefore not a reuse of existing faithfulness tests, but a detection mechanism tailored to explanation attacks that preserve logits.







### To directly address reviewers’ concerns, we also added:





- A fully adaptive, FaithShield-aware attacker that differentiates through Stage I and uses a surrogate for Stage II (Appendix D), showing FaithShield maintains explanation stability and keeps Δconf in the detectable regime even under this stronger threat.





- Cross-backbone transfer experiments (ViT-B/16, B/32, L/14), showing self-attacks reach IoU ≈ 0.44–0.47 while cross-architecture transfer remains strong (e.g., ViT-B/32 → ViT-L/14 IoU ≈ 0.63) with CosSim > 0.94 and MaxΔProb < 4×10⁻⁴.





- Cross-explainer transfer (similarity maps, ScoreCAM, RISE, gradient/IG-style explanations), demonstrating that the vulnerability is representation-level and that Stage I’s robustness transfers to these popular XAI tools.

---

### Note · Authors · 2026-04-08

I have read and agree with the venue's withdrawal policy on behalf of myself and my co-authors.

---

### Meta-Review · Area_Chair_3v61 · 2026-01-07

**Summary:**

This submission studies prediction-preserving explanation manipulation in CLIP-style vision–language models, proposing a new attack (X-Shift) that shifts patch–text explanation maps without changing the final prediction, together with a two-stage defense framework (FaithShield) combining a refined explanation pathway and a masking-based detection mechanism.
The reviewers generally agree that the problem setting—manipulating explanations while preserving predictions—is interesting and potentially relevant to safety-critical deployments. Reviewer GZ9a and UA9t found the formulation technically sound and the experiments extensive, and acknowledged that the rebuttal addressed many clarification and ablation requests.
However, other reviewers (4kAu, RN6w) raised substantial concerns regarding the practical significance of the threat model, limited methodological novelty beyond prior work (notably Li et al.), and whether the proposed attack meaningfully compromises strong or commonly used explanation methods. Despite the additional experiments provided in the rebuttal, these concerns were not fully resolved.
While the paper addresses an interesting and underexplored angle of explanation robustness in VLMs, the practical severity of the demonstrated attack and the distinctiveness of the proposed defense beyond existing work remains unconvinced. The mixed reviewer reception and unresolved concerns make this paper not ready for acceptance this time. I hope the reviewers’ comments can help the authors prepare a better version of this submission.

**Reviewer Concerns:**

Concerns that were addressed in the rebuttal:
- The authors added extensive ablations on the attack loss terms (Eq. 8) and Stage-I components, addressing requests from multiple reviewers.
- Presentation issues (non-vector figures, small fonts, formatting) were clearly fixed.
- Adaptive attacker experiments and cross-backbone / cross-explainer transfer analyses were added and acknowledged by at least one reviewer as satisfactory.
- The relationship between Stage-I and prior work (Li et al.) was clarified more explicitly.

Concerns that remain unsolved:
- Several reviewers remain unconvinced that X-Shift meaningfully manipulates strong, concentrated, widely adopted explanation methods in a way that would realistically mislead users. In particular, the attacked explanations often appear diffuse even before the attack, weakening the argument that a critical failure mode is exposed.
- FaithShield Stage I is widely viewed as a modest extension of Li et al., and Stage II is perceived by multiple reviewers as a variant of existing masking/deletion-based faithfulness tests, despite the authors’ reframing.
- At least one reviewer argues that Stage I alone already yields sufficiently stable explanations, calling into question whether the attack truly necessitates a new defense or detector.
- Despite improved motivation, reviewers remain divided on whether the work justifies its broader claims about safety-critical risks and explanation trustworthiness.

These remaining issues affect both the perceived contribution and the practical relevance of the work.

**Reviewer Scores:**

Based on the discussion and rebuttal:
- Reviewer GZ9a: Likely remains at 6.
- Reviewer UA9t: Likely remains at 6.
- Reviewer 4kAu: Unlikely to increase from 2.
- Reviewer RN6w: Unlikely to increase from 2, and reiterated concerns after rebuttal.

Overall, the score distribution remains bimodal, with strong disagreement and no clear upward convergence.

---

### Decision · Program_Chairs · 2026-01-26

Reject